# Strategically managing learning during perceptual decision making

**Javier Masís[1,2]\*[†], Travis Chapman[2], Juliana Y Rhee[1,2‡], David D Cox[1,2§], Andrew M Saxe[3]\*[#]**

[1]Department of Molecular and Cellular Biology, Harvard University, Cambridge, United States; [2]Center for Brain Science, Harvard University, Cambridge, United States; [3]Department of Experimental Psychology, University of Oxford, Oxford, United Kingdom

**\*For correspondence:**
jmasis@princeton.edu (JM);
a.saxe@ucl.ac.uk (AMS)

**Present address:** [†]Princeton Neuroscience Institute, Princeton University, Princeton, United States; [‡]The Rockefeller University, New York, United States; [§]MIT-IBM Watson AI Lab, Cambridge, United States; [#]Gatsby Unit & Sainsbury Wellcome Centre, University College London, London, United Kingdom

**Competing interest:** The authors declare that no competing interests exist.

**Abstract** Making optimal decisions in the face of noise requires balancing short-term speed and accuracy. But a theory of optimality should account for the fact that short-term speed can influence long-term accuracy through learning. Here, we demonstrate that long-term learning is an important dynamical dimension of the speed-accuracy trade-off. We study learning trajectories in rats and formally characterize these dynamics in a theory expressed as both a recurrent neural network and an analytical extension of the drift-diffusion model that learns over time. The model reveals that choosing suboptimal response times to learn faster sacrifices immediate reward, but can lead to greater total reward. We empirically verify predictions of the theory, including a relationship between stimulus exposure and learning speed, and a modulation of reaction time by future learning prospects. We find that rats' strategies approximately maximize total reward over the full learning epoch, suggesting cognitive control over the learning process.

## Editor's evaluation

This manuscript provides a fresh view on the fundamental trade-off between the speed and accuracy of perceptual decision-making. Using computational modeling, the authors establish the important finding that adopting a momentary suboptimal trade-off for maximizing reward rate at the beginning of learning can yield better decisions and larger rewards at later stages. This novel prediction is tested in rodent experiments. The experiments and their detailed analysis provide compelling evidence for the authors' theoretical predictions.

## Introduction

Optimal behavior in decision making is frequently defined as maximization of reward over time (*Gold and Shadlen, 2002*), and this requires balancing the speed and accuracy of one's choices (*Bogacz et al., 2006*). For example, imagine you are given a multiple-choice quiz on an esoteric topic with which you are familiar, such as behavioral neuroscience or cognitive psychology, and rewarded for every correct answer. In balancing speed and accuracy, you should spend some time on each question to ensure you get it right. Now imagine that you are given a different quiz on an esoteric topic with which you are not familiar, such as low Reynolds number hydrodynamics or underwater basket weaving. In balancing speed and accuracy, you should now guess on as many questions as you can as quickly as you can in order to maximize reward. The ideal balance of speed and accuracy differs considerably in the cases of high and low competence. However, there is an important additional dynamical aspect to consider: competence can change as a function of experience through learning. For instance, taking the hydrodynamics quiz enough times, you might start to get the hang of it,

by going slow enough that you can remember questions and their associated answers, rather than guessing as quickly as you can. Given these almost opposing normative strategies for high and low competence, how does one effectively move from low competence to high competence? In other words, how does an agent strategically manage decision making in light of learning?

In this study, we formalize this problem in the context of a two-choice perceptual decision making task in rodents and simulated agents. Perceptual decisions, in particular two-choice decisions, allow us to leverage one of the most prolific decision making models, the drift-diffusion model (DDM) (*Ratcliff, 1978*) (and the considerable analytical dissections of it *Bogacz et al., 2006*), and one of the most prolific paradigms captured by it, the speed-accuracy trade-off (SAT), as a measurement of optimal behavior (i.e. maximization of reward per unit time) (*Woodworth, 1899*, *Henmon, 1911*, *Garrett, 1922*, *Pew, 1969*, *Pachella, 1974*, *Wickelgren, 1977*, *Ruthruff, 1996*, *Ratcliff and Rouder, 1998*, *Gold and Shadlen, 2002*, *Bogacz et al., 2006*; *Bogacz et al., 2010*; *Heitz and Schall, 2012*; *Heitz, 2014*, *Rahnev and Denison, 2018*).

Studies of the SAT have focused on how the brain may solve it (*Gold and Shadlen, 2002*, *Roitman and Shadlen, 2002*), what the optimal solution is (*Bogacz et al., 2006*), and whether agents can indeed manage it (*Simen et al., 2006*, *Balci et al., 2011a*; *Simen et al., 2009*; *Bogacz et al., 2010*, *Drugowitsch et al., 2014*, *Drugowitsch et al., 2015*; *Manohar et al., 2015*). Though most work in this area has taken place in humans and non-human primates, several studies have established the presence of a SAT in rodents (*Uchida and Mainen, 2003*, *Abraham et al., 2004*; *Rinberg et al., 2006*; *Reinagel, 2013a*; *Reinagel, 2013b*; *Kurylo et al., 2020*). The broad conclusion of much of this literature is that after extensive training, many subjects come close to optimal performance (*Simen et al., 2009Bogacz et al., 2010*; *Balci et al., 2011b*; *Zacksenhouse et al., 2010*, *Balci et al., 2011b*, *Starns and Ratcliff, 2010*, *Holmes and Cohen, 2014*, *Drugowitsch et al., 2014*, *Drugowitsch et al., 2015*). When faced with deviations from optimality, several hypotheses have been proposed, including error avoidance, poor internal estimates of time, and a minimization of the cognitive cost associated with an optimal strategy (*Maddox and Bohil, 1998*, *Bogacz et al., 2006*, *Zacksenhouse et al., 2010*).

Past studies have shown how agents behave after reaching steady-state performance (*Simen et al., 2009*, *Starns and Ratcliff, 2010*, *Bogacz et al., 2010Zacksenhouse et al., 2010*, *Balci et al., 2011b*, *Balci et al., 2011a*; *Starns and Ratcliff, 2010*, *Drugowitsch et al., 2014*, *Drugowitsch et al., 2015*), but relatively less attention has been paid to how agents learn to approach near-optimal behavior (but see *Law and Gold, 2009*, *Balci et al., 2011b*, *Drugowitsch et al., 2019*). While maximizing instantaneous reward rate is a sensible goal when the task is fully mastered, it is less clear that this objective is appropriate during learning.

Here, we set out to understand how agents manage the SAT during learning by studying the learning trajectory of rats and simulated agents in a free-response two-alternative forced-choice visual object recognition task (*Zoccolan et al., 2009*). Rats near-optimally maximized instantaneous reward rate (*iRR*) at the end of learning but chose response times that were too slow to be *iRR*-optimal early in learning. To understand the rats' learning trajectory, we examined learning trajectories in a recurrent neural network (RNN) trained on the same task. We derive a reduction of this RNN to a learning drift-diffusion model (LDDM) with time-varying parameters that describes the network's average learning dynamics. Mathematical analysis of this model reveals a dilemma: at the beginning of learning when error rates are high, *iRR* is maximized by fast responses (*Bogacz et al., 2006*). However, fast responses mean minimal stimulus exposure, little opportunity for perceptual processing, and consequently slow learning. Because of this learning speed/*iRR* (LS/*iRR*) trade-off, slow responses early in learning can yield greater total reward over engagement with the task, suggesting a normative basis for the rats' behavior. We then experimentally tested and confirmed several model predictions by evaluating whether response time and learning speed are causally related, and whether rats choose their response times so as to take advantage of learning opportunities. Our results suggest that rats exhibit cognitive control of the learning process, adapting their behavior to approximately accrue maximal total reward across the entire learning trajectory, and indicate that a policy that prioritizes learning in perceptual tasks may be advantageous from a total reward perspective.

## Results

### Trained rats solve the SAT

We trained $n = 26$ rats on a visual object recognition two-alternative forced-choice task (see Methods) (*Zoccolan et al., 2009*). The rats began a trial by licking the central of three capacitive lick ports, at which time a static visual object that varied in size and rotation from one of two categories appeared on a screen. After evaluating the stimulus, the rats licked the right or left lick port. When correct, they received a water reward, and when incorrect, a timeout period (*Figure 1a*, *Figure 1—figure supplement 1*). Because this was a free-response task, rats were also able to initiate a trial and not make a response, but these ignored trials made up a small fraction of all trials and were not considered during our analysis (*Figure 1—figure supplement 2*).

We examined the relationship between error rate (*ER*) and reaction time (*RT*) during asymptotic performance using the DDM (*Figure 1—figure supplement 3*). In the DDM, perceptual information is integrated through time until the level of evidence for one alternative reaches a threshold. The SAT is controlled by the subject's choice of threshold, and is solved when a subject's performance lies on an optimal performance curve (OPC; *Figure 1b*; *Bogacz et al., 2006*). The OPC defines the mean normalized decision time (*DT*) and *ER* combination for which an agent will collect maximal *iRR* (see Methods). At any given time, an agent will have some perceptual sensitivity (signal-to-noise ratio [SNR]) which reflects how much information about the stimulus arrives per unit time. Given this SNR, an agent's position in speed-accuracy space (the space relating *ER* and *DT*) is constrained to lie on a performance frontier traced out by different thresholds (*Figure 1b*). Using a low threshold yields fast but error-prone responses, while using a high threshold yields slow but accurate responses. An agent only maximizes *iRR* when it chooses the *ER* and *DT* combination on its performance frontier that intersects the OPC. After learning the task to criterion, over half the subjects collected over 99% of their total possible reward, based on inferred SNRs assuming a DDM (*Figure 1c and d*).

Calculating mean normalized *DT* for comparison with the OPC requires knowing two quantities, *DT* and the average non-decision time per error trial $D_{err}$. The average non-decision time $D_{err} = T_0 + \tilde{D}_{err}$ contains the motor and initial perceptual processing components of *RT*, denoted $T_0$; and the post-response timeout on error trials $\tilde{D}_{err}$. Mean normalized *DT* is then the ratio $DT/D_{err}$. In order to determine each subject's *DT*, we estimated $T_0$ through a variety of methods, opting for a biological estimate (measured lickport latency response times and published visual processing latencies; *Figure 1—figure supplement 4*). To ensure that our results did not depend on our choice of $T_0$, we ran a sensitivity analysis on a wide range of possible values of $T_0$ (*Figure 1—figure supplement 4f*). We then had to determine $\tilde{D}_{err}$, which can contain mandatory and voluntary intertrial intervals. We found that the rats generally kept voluntary intertrial intervals to a minimum, and we interpreted longer intervals as effectively 'exiting' the DDM framework (*Figure 1—figure supplement 5*). As such, we defined $\tilde{D}_{err}$ to only contain mandatory intertrial intervals (see Methods, *Figure 1—figure supplement 6*). To ensure that our results did not depend on either choice, we ran a sensitivity analysis on the combined effects of $T_0$ and a $\tilde{D}_{err}$ containing voluntary intertrial intervals on RR (*Figure 1—figure supplement 7*). A full discussion of how these parameters were determined is included in the Methods.

Across a population, a uniform stimulus difficulty will reveal different SNRs because the internal perceptual processing ability in every subject will be different. Thus, although we did not explicitly vary stimulus difficulty (*Simen et al., 2009*, *Bogacz et al., 2010*; *Zacksenhouse et al., 2010Balci et al., 2011b*), as a population, animals clustered along the OPC across a range of *ERs* (*Figure 1d*), supporting the assertion that well-trained rats achieve a near maximal *iRR* in this perceptual task. We note that subjects did not span the entire range of possible *ERs*, and that the differences in optimal *DTs* dictated by the OPC for the *ERs* we did observe are not large. It remains unclear whether our subjects would be optimal over a wider range of task parameters. Notwithstanding, previous work with a similar task found that rats did increase *DTs* in response to increased penalty times, indicating a sensitivity to these parameters (*Reinagel, 2013a*). Thus, for our perceptual task and its parameters, trained rats approximately solve the SAT.

### Rats do not maximize instantaneous reward rate during learning

Knowing that rats harvested reward near-optimally after learning, we next asked whether rats harvested instantaneous reward near-optimally during learning as well. If rats optimized *iRR* throughout learning, their trajectories in speed-accuracy space should always track the OPC.

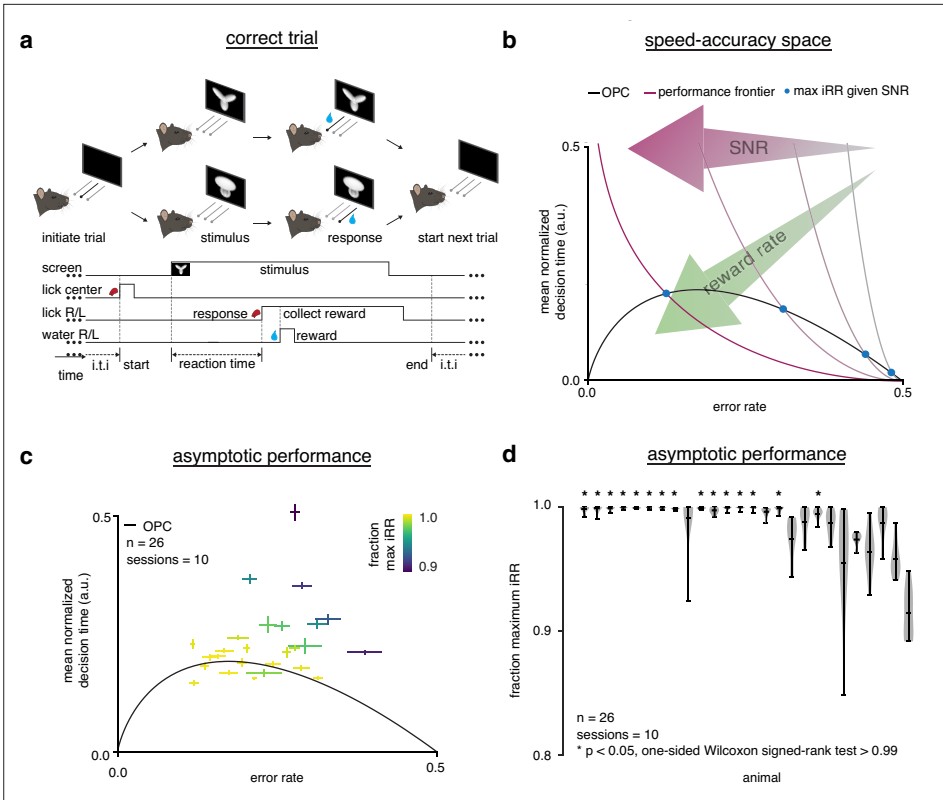

**Figure 1.** Trained rats solve the speed-accuracy trade-off. (**a**) Rat initiates trial by licking center port, one of two visual stimuli appears on the screen, rat chooses correct left/right response port for that stimulus and receives a water reward. (**b**) Speed-accuracy space: a decision making agent's $ER$ and mean normalized $DT$ (a normalization of $DT$ based on the average timing between one trial and the next, see Methods). Assuming a simple drift-diffusion process, agents that maximize $iRR$ (*see* Methods) must lie on an optimal performance curve (OPC, black trace) (*Bogacz et al., 2006*). Points on the OPC relate error rate to mean normalized decision time, where the normalization takes account of task timing parameters (e.g. average response-to-stimulus interval). For a given SNR, an agent's performance must lie on a performance frontier swept out by the set of possible threshold-to-drift ratios and their corresponding error rates and mean normalized decision times. The intersection point between the performance frontier and the OPC is the error rate and mean normalized decision time combination that maximizes $iRR$ for that SNR. Any other point along the performance frontier, whether above or below the OPC, will achieve a suboptimal. $iRR$ Overall, $iRR$ increases toward the bottom left with maximal instantaneous reward rate at error rate = 0.0 and mean normalized decision time = 0.0. (**c**) Mean performance across 10 sessions for trained rats ($n = 26$) at asymptotic performance plotted in speed-accuracy space. Each cross is a different rat. Color indicates fraction of maximum instantaneous reward rate ($iRR$) as determined by each rat's performance frontier. Errors are bootstrapped SEMs. (**d**) Violin plots depicting fraction of maximum, $iRR$ a quantification of distance to the OPC, for same rats and same sessions as c. Fraction of maximum $iRR$ is a comparison of an agent's current $iRR$ with its optimal $iRR$ given its inferred SNR. Approximately 15 of 26 (~60%) of rats attain greater than 99% fraction maximum $iRRs$ for their individual inferred SNRs. * denotes p < 0.05 one-tailed Wilcoxon signed-rank test for mean >0.99.

The online version of this article includes the following figure supplement(s) for figure 1:

**Figure supplement 1.** Task schematic for error trials.

**Figure supplement 2.** Fraction of ignored trials during learning.

**Figure supplement 3.** Drift-diffusion model data fits.

**Figure supplement 4.** Estimating $T_0$.

**Figure supplement 5.** Analysis of voluntary intertrial intervals (ITIs).

**Figure supplement 6.** Mandatory post-error ($\tilde{D}_{err}$) and post-correct ($\tilde{D}_{corr}$) response-to-stimulus interval times.

**Figure supplement 7.** Reward rate sensitivity to $T_0$ and voluntary intertrial interval (ITI).

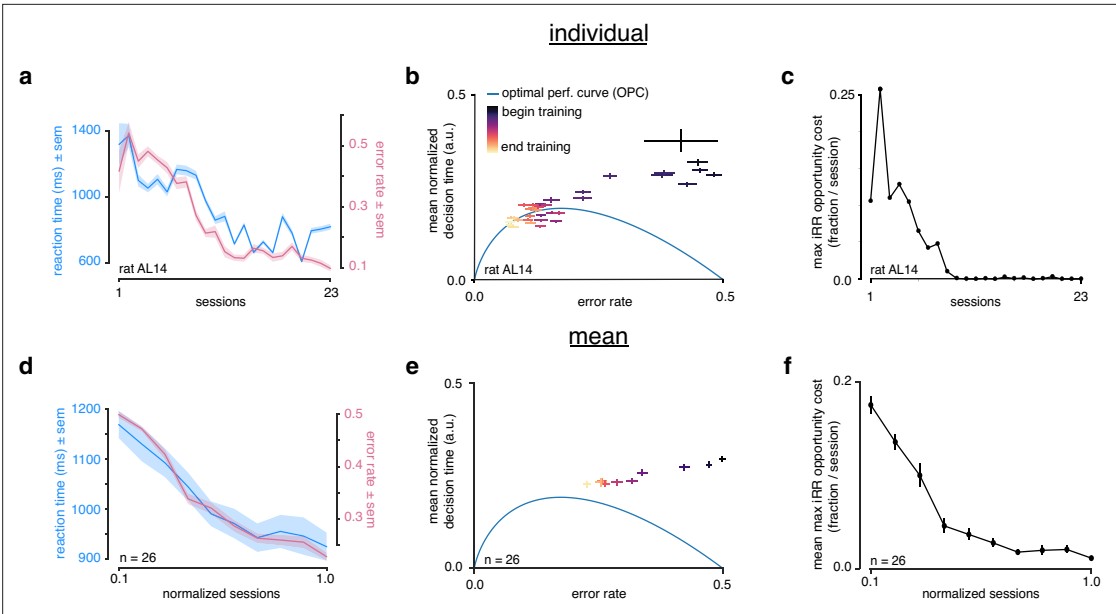

**Figure 2.** Rats do not greedily maximize instantaneous reward rate during learning. (**a**) Reaction time (blue) and error rate (pink) for an example subject (rat AL14) across 23 sessions. (**b**) Learning trajectory of individual subject (rat AL14) in speed-accuracy space. Color map indicates training time. Optimal performance curve (OPC) in blue. (**c**) Maximum *iRR* opportunity cost (see Methods) for individual subject (rat AL14). (**d**) Mean reaction time (blue) and error rate (pink) for $n = 26$ rats during learning. Sessions across subjects were transformed into normalized sessions, averaged and binned to show learning across 10 bins. Normalized training time allows averaging across subjects with different learning rates (see Methods). (**e**) Learning trajectory of $n = 26$ rats in speed-accuracy space. Color map and OPC as in a. (**f**) Maximum *iRR* opportunity cost of rats in b throughout learning. Errors reflect within-subject session SEMs for a and b and across-subject session SEMs for d, e, and f.

The online version of this article includes the following figure supplement(s) for figure 2:

**Figure supplement 1.** Comparison of training regimes.

During learning, a representative individual ($n = 1$) started with long *RTs* that decreased as accuracy increased across training time (***Figure 2a***). Transforming this trajectory to speed-accuracy space revealed that throughout learning the individual did not follow the OPC (***Figure 2b***). Early in learning, the individual started with a much higher *DT* than optimal, but as learning progressed it approached the OPC. The maximum *iRR* opportunity cost is the fraction of maximum possible *iRR* relinquished for a choice of threshold (and average *DT*) (see Methods). We found that this individual gave up over 20% of possible *iRR* at the beginning of learning but harvested reward near-optimally at asymptotic performance (***Figure 2c***). These trends held when the learning trajectories of $n = 26$ individuals were averaged (***Figure 2d–f***). To ensure that our particular training regime (which involved changes in stimulus size and rotation) was not responsible for these trends, we trained a separate cohort ($n = 8$) with a simplified regime that did not involve any changes to the stimuli and we did not observe any meaningful differences (***Figure 2—figure supplement 1***, see Methods). These results show that rats do not greedily maximize *iRR* throughout learning and lead to the question: if rats maximize *iRR* at the end of learning, what principle governs their strategy at the beginning of learning?

## Learning DDM

To theoretically understand the effect of different learning strategies, we developed a simple linear RNN formalism for our task. This framework enables investigation of how long-term perceptual learning across many trials is influenced by the choice of decision time on individual trials (***Figure 3***). We first describe this neural network formalism, before showing how it can be analytically reduced to a classic DDM with time-dependent parameters that evolve over the course of learning.

### Linear RNN

Our model takes the form of a simple RNN, depicted unrolled through time in ***Figure 3a***. The network receives noisy sensory input over time during a trial, amplifies this evidence through weighted

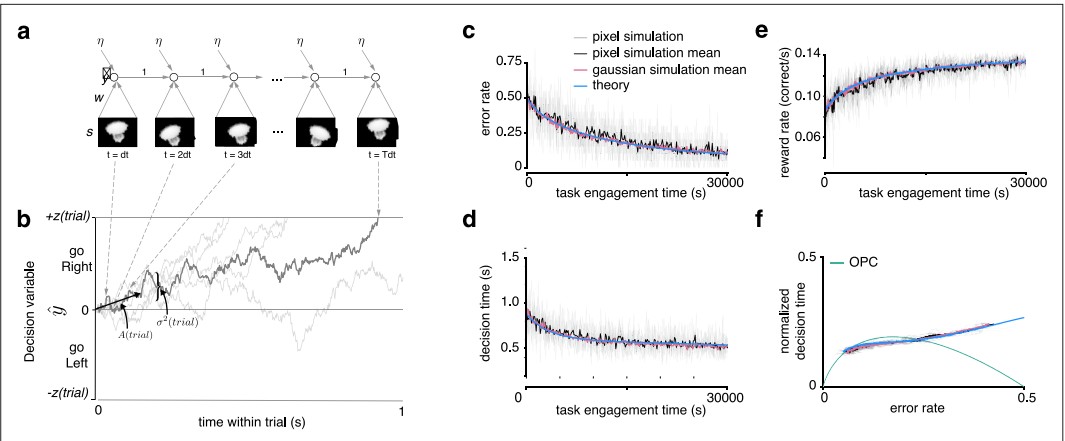

**Figure 3.** Recurrent neural network and learning drift diffusion model (DDM). (**a**) Roll out in time of recurrent neural network (RNN) for one trial. (**b**) The decision variable for the recurrent neural network (dark gray), and other trajectories of the equivalent DDM for different diffusion noise samples (light gray). (**c, d, e**) Changes in $ER$, $DT$, and $iRR$ over a long period of task engagement in the RNN (light gray, pixel simulation individual traces; black, pixel simulation mean; pink, Gaussian simulation mean) compared to the theoretical predictions from the learning DDM (blue). (**f**) Visualization of traces in c and d in speed-accuracy space along with the optimal performance curve (OPC) in green. The threshold policy was set to be $iRR$-sensitive for c–f.

The online version of this article includes the following figure supplement(s) for figure 3:

**Figure supplement 1.** Analytical reduction of linear drift-diffusion model (LDDM) matches error-corrective learning neural network dynamics during learning.

synaptic connections, and integrates the result until a threshold is reached. After making a decision and receiving feedback, the synaptic connections are updated a small amount according to an error-corrective gradient descent learning rule. Therefore, there are two key timescales in the model: first, the fast activity dynamics during a single trial, which produces a single decision with a certain reaction time; and second, the slow weight dynamics due to learning across many trials. In the following, we denote time within trial as the variable $t$, and the trial number as $trial$. We now describe the dynamics on each timescale in greater detail.

Within a trial, $N$ dimensional inputs $s(t) \in R^N$ arrive at discrete times $t = 1dt, 2dt, \cdots$, where $dt$ is a small time step parameter. In our experimental task, $s(t)$ might represent the activity of LGN neurons in response to a given visual stimulus. Because of eye motion and noise in the transduction from light intensity to visual activity, the response of individual neurons will only probabilistically relate to the correct answer at any given instant. In our simulations, we take $s(t)$ to be the pixel values of the exact images presented to the animals, but transformed at each time point by small rotations (±20°) and translations (±25% of the image width and height), as depicted in *Figure 3a*. This input variability over time makes temporal integration valuable even in this visual classification task. To perform this integration, each input $s(t)$ is filtered through perceptual weights $w(trial) \in R^N$ and added to a read-out node (decision variable) $\hat{y}(t)$ along with i.i.d. integrator noise $\eta(t) \sim \mathcal{N}(0, c_o^2 dt)$. This integrator noise models internal neural noise. The evolution of the decision variable is given by the simple linear recurrence

$$\hat{y}(t + dt) = \hat{y}(t) + w(trial) \cdot s(t) + \eta(t), \tag{1}$$

until the decision variable hits a threshold $\pm z(trial)$ that is constant on each trial. Here, the RNN already performs an integration through time (a choice motivated by prior experiments in rodents *Brunton et al., 2013*), and improvements in performance come from adjusting the input-to-integrator weights $w(trial)$ to better extract task-relevant sensory information.

Across trials, the perceptual weights $w(trial)$ are updated to improve performance. In principle this could be accomplished with many possible learning mechanisms such as reinforcement learning (*Law and Gold, 2009*) or Bayesian inference (*Drugowitsch et al., 2019*). Here, we investigate gradient-based optimization of an objective function, as commonly used in deep learning approaches (*Richards*

et al., 2019, **Saxe et al., 2021**). In particular, we consider using gradient descent on the hinge loss, corresponding to standard practice in deep learning. The hinge loss is

$$Loss(trial) = \max(0, 1 - y(trial)\hat{y}(trial)) \tag{2}$$

where $y(trial) = \pm 1$ is the correct output sign for the trial. Then the weights are updated by gradient descent on this loss,

$$w(trial + 1) = w(trial) - \lambda \frac{\partial Loss(trial)}{\partial w}, \tag{3}$$

where $\lambda$ is a small learning rate. The hinge loss is a proxy for accuracy, and so this weight update implements a learning scheme based on error feedback. In essence, perceptual weights are updated after error trials to improve the likelihood of answering correctly in the future.

To summarize the key parameters of the RNN, the model requires specifying the input distribution $s(t)$, the initial perceptual weights $w(0)$, the integrator noise variance $c_o^2$, the gradient descent learning rate $\lambda$, and the decision threshold $z(trial)$ used on each trial. With these parameters specified, the model can be simulated to make predictions for how behavior will evolve over training, as shown in **Figure 3c–f**, gray and black traces.

## Reduction to LDDM

While the behavior of the RNN model obtained in simulations can be compared to data, deep network models remain challenging to understand (**Saxe et al., 2021**). We therefore sought to mathematically analyze this setting to derive a simple theory of the average learning dynamics that highlights key trade-offs.

We start by noting that the input to the decision variable $\hat{y}$ at each time step is a weighted sum of many random variables, which by the law of large numbers will be approximately Gaussian. We therefore develop a reduction of this model based on an effective Gaussian scalar input distribution. At each time step the input pathway receives a Gaussian input $x(t) \sim \mathcal{N}(Aydt, c_i^2 dt)$, where $A$ parametrizes the signal related to $y$, and the input noise variance $c_i^2$ parametrizes irreducible noise in input channels that cannot be rejected. This input is multiplied by a scalar weight $u$, added to output noise $\eta$ of variance $c_o^2$ and sent into the integrating node $\hat{y}$,

$$\hat{y}(t + dt) = \hat{y}(t) + u(trial)x(t) + \eta(t), \tag{4}$$

where we emphasize that $u$ and $x(t)$ are now both scalar. We may then perform gradient descent on the hinge loss, yielding the update $u(trial + 1) = u(trial) - \lambda \frac{\partial Loss(trial)}{\partial u}$. As expected from the law of large numbers, for the right choice of input signal and parameters $A$ and $c_i$, simulations of this effective Gaussian model closely match the full simulation from pixels, as shown in **Figure 3c–f**, pink trace.

Next, to relate these dynamics to the well-studied DDM framework, we examine behavior when the time step is small ($dt \to 0$) to obtain a continuous time formulation. In the continuum limit, these discrete within-trial dynamics of the network yield decision variables with identical distributions to a drift-diffusion process with an effective SNR $\bar{A}$ and normalized threshold $\bar{z}$

$$\bar{A} = \frac{A^2 u^2}{u^2 c_i^2 + c_o^2}, \tag{5}$$

$$\bar{z} = \frac{z}{Au}, \tag{6}$$

yielding the mean error rate (ER) and decision time (DT)

$$ER = \frac{1}{1 + e^{2\bar{z}\bar{A}}}, \tag{7}$$

$$DT = \bar{z}\tanh\left(\bar{z}\bar{A}\right). \tag{8}$$

Finally, we assume that the learning rate is small ($\lambda \ll 1$), such that weights change little on any given trial and the gradient dynamics are driven by the mean update,

$$u(trial + dt) = u(trial) - \lambda \left\langle \frac{\partial Loss(trial)}{\partial u} \right\rangle, \tag{9}$$

where $\langle \cdot \rangle$ denotes the average with respect to the distribution of outputs obtained with perceptual weights $u(trial)$ and threshold $z(trial)$. These average dynamics depend in a complex way on the current performance of the network. We compute these average dynamics analytically (see Methods), yielding the continuous time change in effective SNR in the DDM that is equivalent to gradient descent learning in the underlying neural network model. In particular, gradient descent in the RNN is equivalent to the following SNR dynamics in the DDM:

$$\tilde{\tau} \frac{d}{dt} \bar{A}(t) = 2 \sqrt{\frac{\bar{A}(t) \left( \bar{A}^* \right)}{c}} \left( 1 - \frac{\bar{A}(t)}{\bar{A}^*} \right)^{5/2} \frac{ER(t)}{DT(t) + D_{tot}(t)} \left[ DT(t) - \frac{\log(1/ER(t) - 1)}{\bar{A}^* \left( 1 - \frac{\bar{A}(t)}{\bar{A}^*} \right)^2} \right]. \tag{10}$$

Here, time $t$ measures seconds of task engagement (i.e. it measures time passing within a trial as well as intertrial time and any penalty delays after error trials), and $D_{tot}(t) = (1 - ER(t))D_{corr} + (ER(t))D_{err}$ is the average non-decision task engagement time per trial (where $D_{corr}$ and $D_{err}$ are the average non-decision task engagement times after correct and error trials). The SNR dynamics depend on five parameters: the time constant $\tilde{\tau}$ related to the learning rate, the initial SNR $\bar{A}(0)$, the asymptotic achievable SNR after learning $\bar{A}^*$, the integration-noise to input-noise variance ratio $c \equiv c_o^2/c_i^2$, and the choice of threshold $z(t)$ over training. We note that the dependence of the dynamics on the choice of threshold $z(t)$ is implicit in $ER(t), DT(t)$, and $D_{tot}(t)$ in **Equation 10**. The dynamics of this LDDM closely tracks simulated trajectories of the full network from pixels (**Figure 3c–f** blue trace, **Figure 3—figure supplement 1**; see Methods).

Remarkably, this reduction shows that the high-dimensional dynamics of the RNN receiving stochastic pixel input and performing gradient descent on the weights (**Figure 3**, gray trace) can be described by a DDM with a single deterministic scalar variable – the effective SNR – that changes over time (**Figure 3**, blue trace). Notably, without the mapping to the original RNN, it is not possible to understand what effect error-corrective gradient descent learning would have at the level of the DDM, or how the learning process is influenced by choice of decision times. In particular, the change in SNR that arises from gradient descent on the underlying RNN weights (**Equation 10**) is not equivalent to that arising from gradient descent on the SNR parameter in the DDM directly because gradient descent is not parametrization invariant.

## Learning speed trades off with instantaneous reward rate

The LDDM reveals that learning dynamics depend on the choice of threshold $z(t)$ on each trial over learning, because threshold impacts both error rate and decision time, which appear in the SNR dynamics of **Equation 10**. We next sought to qualitatively understand this relationship. A key prediction of the LDDM is a tension between learning speed and $iRR$, the LS/$iRR$ trade-off. This tension is clearest early in learning when $ERs$ are near 50%. Then the rate of change in SNR is

$$\frac{d}{dt} \bar{A} \propto \frac{DT}{DT + D_{tot}}, \tag{11}$$

where the proportionality constant does not depend on $DT$ (see derivation, Methods). Hence learning speed increases with increasing $DT$. By contrast, when accuracy is 50% the $iRR$ decreases with increasing $DT$,

$$iRR(t) \approx \frac{1/2}{DT + D_{tot}}. \tag{12}$$

When encountering a new task, therefore, agents face a dilemma: they can either harvest a large $iRR$ or they can learn quickly.

## Learning dynamics depend on threshold policies

Just as the standard DDM instantiates different decision making strategies as different choices of threshold (for instance aimed at maximizing $iRR$, accuracy, or robustness) (**Holmes and Cohen, 2014**;

*Zacksenhouse et al., 2010*), the LDDM instantiates different learning strategies through the choice of threshold trajectory over learning. Threshold affects $DT$ and $ER$, and through these, the learning dynamics in *Equation 10*. To consider a range of strategies, we developed four potential threshold policies.

Constant threshold. This policy implements a fixed constant threshold $z^c(t) = z_0$. It serves as a control for behavior that would arise without the ability to modulate decision threshold. Constant thresholds across difficulties have been found to be used as part of near-optimal and presumably cognitively cheaper strategies in humans (*Balci et al., 2011b*). This policy introduces the parameter $z_0$.

*iRR*-greedy. This policy sets the threshold to the value that maximizes instantaneous reward on each trial, $z^g(t) = z^*(\bar{A})$, such that behavior always lies on the OPC. This instantiates a 'myopic' strategy that does not consider how threshold can impact long-term learning. This policy is similar to a previously proposed neural network model of rapid threshold adjustment based on reward rate (*Simen et al., 2006*). The policy introduces no parameters.

*iRR*-sensitive. This policy implements a threshold $z^s(t)$ that decays with time constant $\gamma$ from an initial value $z^s(0) = z_0$ toward the *iRR*-optimal threshold,

$$\gamma \frac{d}{dt} z^s(t) = z^*(\bar{A}(t)) - z^s(t).$$

Notably, as the SNR changes due to learning, the target threshold also changes through time. Asymptotically, this policy converges to greedy *iRR*-optimal behavior; however, by starting with a high initial threshold, it can undergo a transient period where responses are slower or faster than *iRR*-optimal, potentially influencing learning. It instantiates a heuristic strategy in which behavior differs from *iRR*-optimal behavior early in learning. This policy introduces two parameters, $z_0$ and $\gamma$.

Global optimal. This policy selects the threshold $z^o(t)$ that maximizes total cumulative reward at some known predetermined end to the task $T_{tot}$,

$$z^o(t) = \underset{z(t)}{\mathrm{argmax}} \int_0^{T_{tot}} RR(t) dt.$$

We approximately compute this threshold function using automatic differentiation (see Methods). This policy serves as a normative oracle to which behavior may be compared. We note that this optimal policy considers the full time course of learning and is aware of all task parameters such as the duration of total task engagement $T_{tot}$, asymptotically achievable SNR $A^*$, etc. In practice these parameters cannot be known before experiencing the task, and so this policy is not an implementable strategy but a normative reference point. The policy introduces no parameters.

In designing this model, we kept components as simple as possible to highlight key qualitative trade-offs between learning speed and decision strategy. Because of its simplicity, like the standard DDM, it is not meant to quantitatively describe all aspects of behavior. We instead use it to investigate qualitative features of decision making strategy, and expect that these features would be preserved in other related models of perceptual decision making (*Usher and McClelland, 2001Mazurek et al., 2003, Gold and Shadlen, 2007, Heekeren et al., 2004Heekeren et al., 2008Ma et al., 2006Brown and Heathcote, 2008, Ratcliff and McKoon, 2008, Beck et al., 2008, Roitman and Shadlen, 2002; Purcell et al., 2010Bejjanki et al., 2011; Drugowitsch et al., 2012; Fard et al., 2017*).

## Model reveals that prioritizing learning can maximize total reward

In order to qualitatively understand how these models behave through time, we visualized their learning dynamics. To approximately place the LDDM task parameters in a similar space to the rats, we performed maximum likelihood fitting using automatic differentiation through the discretized reduction dynamics (see Methods). The four policies we considered clustered into two groups, distinguished by their behavior early in learning. A 'greedy' group, which contained just the *iRR*-greedy policy, remained always on the OPC (*Figure 4a*), and had fast initial response times (*Figure 4b*), a long initial period at high error (*Figure 4c*), and high initial *iRR* (*Figure 4d*). By contrast, a 'non-greedy' group, which contained the *iRR*-sensitive, constant threshold, and global optimal policies, started far above the OPC (*Figure 4a*), and had slow initial response times (*Figure 4b*), rapid improvements in ER (*Figure 4c*), and low *iRR* (*Figure 4d*). Notably, while members of the non-greedy group started off with lower *iRR*, they rapidly surpassed the slow learning group (*Figure 4d*) and ultimately accrued

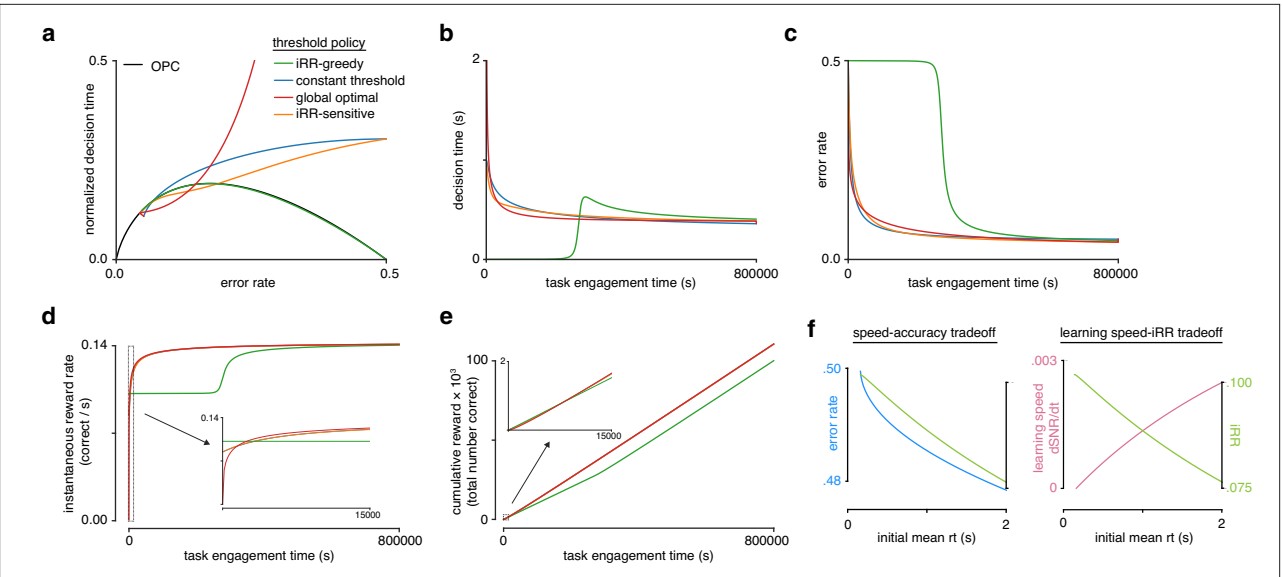

**Figure 4.** Model reveals rat learning dynamics lead to higher instantaneous reward rate and long-term rewards than greedily maximizing instantaneous reward rate. (**a**) Model learning trajectories in speed-accuracy space plotted against the optimal performance curve (OPC) (black). (**b**) Decision time through learning for the four different threshold policies in a. (**c**) Error rate throughout learning for the four different threshold policies in a. (**d**) Instantaneous reward rate as a function of task engagement time for the full learning trajectory and a zoom-in on the beginning of learning (*inset*). (**e**) Cumulative reward as a function of task engagement time for the full learning trajectory and a zoom-in on the beginning of learning (*inset*). Threshold policies: *iRR*-greedy (green), constant threshold (blue), *iRR*-sensitive (orange), and global optimal (red). (**f**) In the speed-accuracy trade-off (*left*), *ER* (blue) decreases with increasing initial mean *RT* (green) at high error rates (~0.5) also decreases with increasing initial mean *RT*. Thus, at high *ERs*, an agent solves the speed-accuracy trade-off by choosing fast *RTs* that result in higher *ERs* and maximize *iRR*. In the learning speed/ *iRR* trade-off (*right*), initial learning speed (*dSNR/dt*, pink) increases with increasing initial mean *RT*, whereas *iRR* (green) follows the opposite trend. Thus, an agent must trade *iRR* in order to access higher learning speeds. Plots generated using linear drift-diffusion model (LDDM).

The online version of this article includes the following figure supplement(s) for figure 4:

**Figure supplement 1.** Allowing both drift rate and threshold to vary with learning provides the best drift-diffusion model (DDM) fits.

**Figure supplement 2.** Simple drift-diffusion model (DDM) fits indicate threshold decreases and drift rate increases during learning.

**Figure supplement 3.** Simple drift-diffusion model (DDM) + fixed drift rate variability fits indicate threshold decreases and drift rate increases during learning.

**Figure supplement 4.** Simple drift-diffusion model (DDM) + variable drift rate variability fits indicate threshold decreases and drift rate increases during learning.

**Figure supplement 5.** Model reveals rat learning dynamics resemble optimal trajectory without relinquishing initial rewards.

more total reward (*Figure 4e*). Overall, these results show that threshold strategy strongly impacts learning dynamics due to the learning speed/*iRR* trade-off (*Figure 4f*), and that prioritizing learning speed can achieve higher cumulative reward than prioritizing instantaneous reward rate.

We further analyzed the differences between the three strategies in the non-greedy group. The global optimal policy selects extremely slow initial *DTs* to maximize the initial speed of learning. By contrast, the *iRR*-sensitive and constant threshold policies start with moderately slow responses. Nevertheless, we found that these simple strategies accrued 99% of the total reward of the global optimal strategy (*Figure 4—figure supplement 5*). Hence these more moderate policies, which do not require oracle knowledge of future task parameters, derive most of the benefit in terms of total reward and may reflect a reasonable approach when the duration of task engagement is unknown.

Considering the rats' trajectories in light of these strategies, their slow responses early in learning stand in stark contrast to the fast responses of the *iRR*-greedy policy (*Figure 2b*, *Figure 4a*). Equally, their responses were faster than the extremely slow initial *DTs* of the global optimal model. Both the *iRR*-sensitive and constant threshold models qualitatively matched the rats' learning trajectory. However, the best DDM parameter fits of the rats' behavior allowed their thresholds to decrease throughout learning, failing to support the constant threshold model (*Figure 4—figure supplements 1–4*). Subsequent experiments (Figure 6) provide further evidence against a simple constant threshold

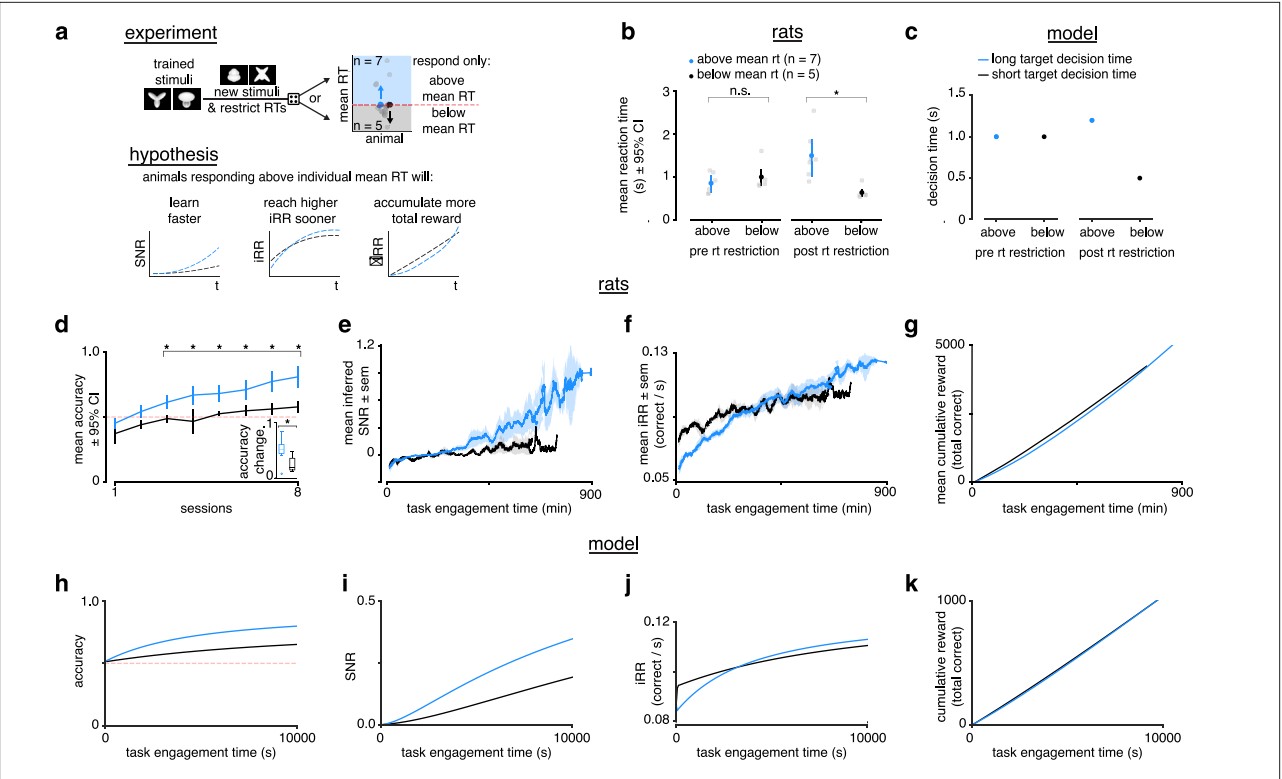

**Figure 5.** Longer reaction times lead to faster learning and higher instantaneous reward rates. (**a**) Schematic of experiment and hypothesized results. Previously trained animals were randomly divided into two groups: could only respond above (blue, $n = 7$) or below (black, $n = 5$) their individual mean reaction times for the previously trained stimulus and the new stimulus. Subjects responding above their individual mean reaction times were predicted to learn faster, reach a higher instantaneous reward rate sooner and accumulate more total reward. (**b**) Mean and individual reaction times before and after the reaction time restriction in rats. The mean reaction time for subjects randomly chosen to respond above their individual mean reaction times (blue, $n = 7$) was not significantly different to those randomly chosen to respond below their individual means (black, $n = 5$) before the restriction (Wilcoxon rank-sum test p > 0.05), but were significant after the restriction (Wilcoxon rank-sum test p < 0.05). Errors represent 95% confidence intervals. (**c**) In the model a long (blue) and a short (black) target decision time were set through a control feedback loop on the threshold, $\frac{d}{dt}z(t) = \gamma(DT_{targ} - DT(t))$ with parameter $\gamma = 0.01$. (**d**) Mean accuracy ±95% confidence interval across sessions for rats required to respond above (blue, $n = 7$) or below (black, $n = 5$) their individual mean reaction times for a previously trained stimulus. Both groups had initial accuracy below chance because rats assume a response mapping based on an internal assessment of similarity of new stimuli to previously trained stimuli. To counteract this tendency and ensure learning, we chose the response mapping for new stimuli that contradicted the rats' mapping assumption, having the effect of below-chance accuracy at first. * denotes p < 0.05 in two-sample independent *t*-test. *Inset*: accuracy change (slope of linear fit to accuracy across sessions to both groups, units: fraction per session). * denotes p < 0.05 in a Wilcoxon rank-sum test. (**e**) Mean inferred signal-to-noise ratio (SNR), (**f**) mean, *iRR* and (**g**) mean cumulative reward across task engagement time for new stimulus pair for animals in each group. (**h**) Accuracy, (**i**) SNR, (**j**) *iRR*, and (**k**) cumulative reward across task engagement time for long (blue) and short (black) target decision times in the linear drift-diffusion model (LDDM).

strategy. Consistent with substantial improvements in perceptual sensitivity through learning, DDM fits to the rats also showed an increase in drift rate throughout learning (*Figure 4—figure supplements 1–4*). Similar increases in drift rate have been observed as a universal feature of learning throughout numerous studies fitting learning data with the DDM (*Ratcliff et al., 2006*; *Dutilh et al., 2009Petrov et al., 2011Balci et al., 2011b*, *Liu and Watanabe, 2012*, *Zhang and Rowe, 2014*). These qualitative comparisons suggest that rats adopt a 'non-greedy' strategy that trades initial rewards to prioritize learning in order to harvest a higher *iRR* sooner and accrue more total reward over the course of learning.

## Learning speed scales with reaction time

To test the central prediction of the LDDM that learning (change in SNR) scales with mean $DT$, we designed an $RT$ restriction experiment and studied the effects of the restriction on learning in the rats. Previously trained rats ($n = 12$) were randomly divided into two groups in which they would have to learn a new stimulus pair while responding above or below their individual mean $RTs$ ('slow'

and 'fast') for the previously trained stimulus pair (*Figure 5a*). Before introducing the new stimuli, we carried out practice sessions with the new timing restrictions to reduce potential effects related to a lack of familiarity with the new regime. After the restriction, *RTs* were significantly different between the two groups (*Figure 5b*). In the model, we simulated an *RT* restriction by setting two different *DTs* (*Figure 5c*).

We found no difference in initial mean session accuracy between the two groups, followed by significantly higher accuracy in the slow group in subsequent sessions (*Figure 5d*). The slope of accuracy across sessions was significantly higher in the slow group (*Figure 5d*, inset). Importantly, the fast group had a positive slope and an accuracy above chance by the last session of the experiment, indicating this group learned (*Figure 5d*).

Because of the SAT in the DDM, however, accuracy could be higher in the slow group even with no difference in perceptual sensitivity (SNR) or learning speed simply because on average they view the stimulus for longer during a trial, reflecting a higher threshold. To see if underlying perceptual sensitivity increased faster in the slow group, we computed the rats' inferred SNR throughout learning (see Methods, *Equation 24*), which takes account of the relationship between *RT* and *ER*. The SNR of the slow group increased faster (*Figure 5e*), consistent with a learning speed that scales with *DT*.

We found that the slow group had a lower initial *iRR*, but that this *iRR* exceeded that of the fast group halfway through the experiment (*Figure 5f*). Similarly, the slow group trended toward a higher cumulative reward by the end of the experiment (*Figure 5g*). The LDDM qualitatively replicates all of our behavioral findings (*Figure 5h–k*). These results demonstrate the potential total reward benefit of faster learning, which in this case was a product of enforced slower *RTs*.

Our experiments and simulations demonstrate that longer *RTs* lead to faster learning and higher reward for our task setting both in vivo and in silico. Moreover, they are consistent with the hypothesis that rats choose high initial *RTs* in order to prioritize learning and achieve higher *iRRs* and cumulative rewards during the task.

## Rats choose reaction time based on learning prospects

The previous experiments suggest that rats trade initial rewards for faster learning. Nonetheless, it is unclear how much control rats exert over their *RTs*. A control-free heuristic approach, such as adopting a fixed high threshold (our constant threshold policy), might incidentally appear near optimal for our particular task parameters, but might not be responsive to changed task conditions. If an agent is controlling the reward investment it makes in the service of learning, then it should only make that investment if it is possible to learn.

To test whether the rats' *RT* modulations were sensitive to learnability, we conducted a new experiment in which we divided rats into a group that encountered new learnable visible stimuli ($n = 16$, sessions = 13), and another that encountered unlearnable transparent or near-transparent stimuli ($n = 8$, sessions = 11) (*Figure 6a*). From the perspective of the LDDM, both groups start with approximately zero SNR, however only the group with the visible stimuli can improve that SNR. Because the rats do not know the learnability of new stimuli, we initialize the LDDM with a high threshold to model the belief that any new stimuli may be learnable. If the rats choose their *RTs* based on how much it is possible to learn, then: (1) rats encountering new stimuli that they can learn will increase their *RTs* to learn quickly and increase future *iRR*. (2) Rats encountering new stimuli that they cannot learn might first increase their *RTs* to learn that there is nothing to learn, but (3) will subsequently decrease *RTs* to maximize *iRR*.

We found that the rats with the visible stimuli qualitatively replicated the same trajectory in speed-accuracy space that we found when rats were trained for the first time (*Figure 2b*, *Figure 6b*). Indeed, the best DDM fits were those that allowed both threshold and drift rate to vary with learning, as was the case with the first stimuli the rats encountered, and in line with the LDDM (*Figure 4—figure supplements 1–4*). Because these previously trained rats had already mastered the task mechanics, this result rules out non-stimulus-related learning effects as the sole explanation for long *RTs* at the beginning of learning and supports our hypothesis that the slowdown in *RT* was attributable to the rats trying to learn the new stimuli efficiently. We calculated the mean change in *RT* (mean Δ*RT*) of new stimuli versus known stimuli. The visible stimuli group had a significant slowdown in *RT* lasting many sessions that returned to baseline by the end of the experiment (*Figure 6d*, black trace).

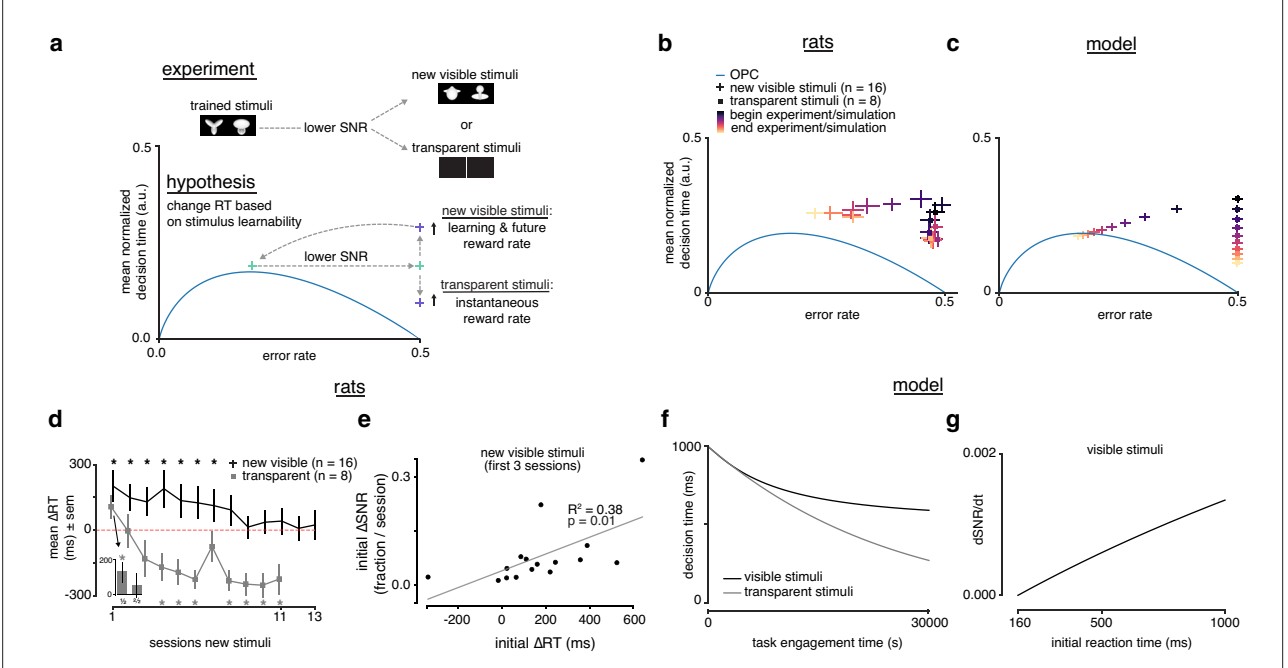

**Figure 6.** Rats choose reaction time based on stimulus learnability. (**a**) Schematic of experiment: rats trained on stimulus pair 1 were presented with new visible stimulus pair 2 or transparent (alpha = 0, 0.1) stimuli. If rats change their reaction times based on stimulus learnability, they should increase their reaction times for the new visible stimuli to increase learning and future *iRR* and decrease their reaction time to increase *iRR* for the transparent stimuli. (**b**) Learning across normalized sessions in speed-accuracy space for new visible stimuli ($n = 16$, crosses) and transparent stimuli ($n = 8$, squares). Color map indicates time relative to start and end of the experiment. (**c**) *iRR*-sensitive threshold model runs with 'visible' (crosses) and 'transparent' (squares) stimuli (modeled as containing some signal, and no signal) plotted in speed-accuracy space. The crosses are illustrative and do not reflect any uncertainty. Color map indicates time relative to start and end of simulation. (**d**) Mean change in reaction time across sessions for visible stimuli or transparent stimuli compared to previously known stimuli. Positive change means an increase relative to previous average. *Inset*: first and second half of first session for transparent stimuli. * denotes $p < 0.05$ in permutation test. (**e**) Correlation between initial individual mean change in reaction time (quantity in d) and change in signal-to-noise ratio (SNR) (learning speed: slope of linear fit to SNR per session) for first three sessions with new visible stimuli. $R^2$ and $p$ from linear regression in d. Error bars reflect standard error of the mean in b and d. (**f**) Decision time across time engagement time for visible and transparent stimuli runs in model simulation. (**g**) Instantaneous change in SNR ($\frac{d}{dt}\bar{A}$) as a function of initial reaction time (decision time + non-decision time $T_0$) in model simulation.

The online version of this article includes the following figure supplement(s) for figure 6:

**Figure supplement 1.** Reaction time analysis of transparent stimuli experiment.

**Figure supplement 2.** Vincentized reaction time distributions throughout learning.

**Figure supplement 3.** Simple drift-diffusion model (DDM) + variable drift rate variability fits for transparent stimuli.

**Figure supplement 4.** Analysis of stimulus-independent strategies for transparent stimuli.

**Figure supplement 5.** Post-error slowing during rat learning dynamics.

Rats with the transparent stimuli also approached the OPC by decreasing their *RTs* across sessions to better maximize *iRR* (**Figure 6b**). After a brief initial increase in *RT* in the first half of the first session (**Figure 6d**, inset), *RTs* rapidly decreased (**Figure 6d**, gray trace). Notably, *RTs* fell below the baseline *RTs*, indicating a strategy of responding quickly, which approaches *iRR*-optimal behavior for this zero SNR task. Additionally, we considered the rats' entire *RT* distributions to investigate the effect of learnability beyond *RT* means. We found that while the *RT* distributions changed similarly from the beginning to end of learning for the learnable stimuli (stimulus pair 1 and 2), they differed for the unlearnable (transparent) stimuli, indicating an effect of learnability on the entire *RT* distributions (**Figure 6—figure supplement 2**). Hence, rodents are capable of modulating their strategy depending on their learning prospects.

Although there is no informative signal in this task with transparent stimuli, the rats could still be using stimulus-independent signals, such as choice history or feedback, to drive heuristic strategies. Indeed, DDM fits indicated a non-zero drift rate even in the absence of informative stimuli

(**Figure 6—figure supplement 3**). To investigate whether the rats implemented stimulus-independent heuristic strategies in addition to random choice, we measured left/right bias and quantified the weights of bias, perseverance (choose the same port as the previous trial), and win-stay/lose-switch (choose the port that was correct on the previous trial) (**Roy et al., 2021**). In general, bias seemed to increase with transparent stimuli in the direction that each individual was already biased during visible stimuli. Perseverance and win-stay/lose-switch also seemed to increase and fluctuate more during transparent stimuli, suggesting a greater reliance on these heuristics now that the stimulus was uninformative (**Figure 6—figure supplement 4**). Engaging these heuristics may be a way that the rats expedited their choices in order to maximize *iRR* while still 'monitoring' the task for any potentially informative changes or patterns. Despite the fact that the animals' still engaged these non-optimal heuristics, the lack of learnability in the transparent stimuli still led to a change in strategy that was distinct from that with learnable stimuli.

Importantly, this learnability experiment argues against other simple strategies accounting for the changes in *RTs*. If rats respond more slowly after error trials, a phenomenon known as post-error slowing (PES), they might exhibit slower *RTs* early in learning when errors are frequent (**Notebaert et al., 2009**). Indeed, we found a slight mean post-error slowing effect of about 50 ms that was on average constant throughout learning, though it was highly variable across individuals (**Figure 6—figure supplement 5**). However, rats viewing transparent stimuli had *ERs* constrained to 50%, yet their *RTs* systematically decreased (**Figure 6b**), such that post-error slowing alone cannot account for their strategy. Similarly, choosing *RTs* as a simple function of time since encountering a task would not explain the difference in *RT* trajectories between visible and transparent stimuli (**Figure 6d**).

A simulation of this experiment with the *iRR*-sensitive threshold LDDM qualitatively replicated the rats' behavior (**Figure 6c, f and g**). Rodent behavior is thus consistent with a threshold policy that starts with a relatively long *DT* upon encountering a new task, and then decays toward the *iRR*-optimal *DT*. All other threshold strategies we considered fail to account for the totality of the results. The *iRR*-greedy strategy – as before – stays pinned to the OPC and speeds up upon encountering the novel stimuli rather than slowing down. The constant threshold strategy fails to predict the speed-up in *DT* for the transparent stimuli if we assume constant diffusion noise. This is because when the perceptual signal is small, mean *DT* can be shown to be the squared ratio of threshold to diffusion noise (see Methods). It is thus also possible to explain the speed-up with a constant threshold and increasing diffusion noise. With either interpretation, however, it is clear that a policy where the ratio of threshold to diffusion noise is constant is not compatible with the results. Finally, the global optimal strategy (which has oracle knowledge of the prospects for learning in each task) behaves like the *iRR*-greedy policy from the start on the transparent stimuli as there is nothing to learn.

Our *RT* restriction experiment showed that higher initial *RTs* led to faster learning, a higher *iRR* and more cumulative reward. Consistent with these findings, there was a correlation between initial mean $\Delta RT$ and initial $\Delta SNR$ across subjects viewing the visible stimuli, indicating the more an animal slowed down, the faster it learned (**Figure 6e**). We further tested these results in the voluntary setting by

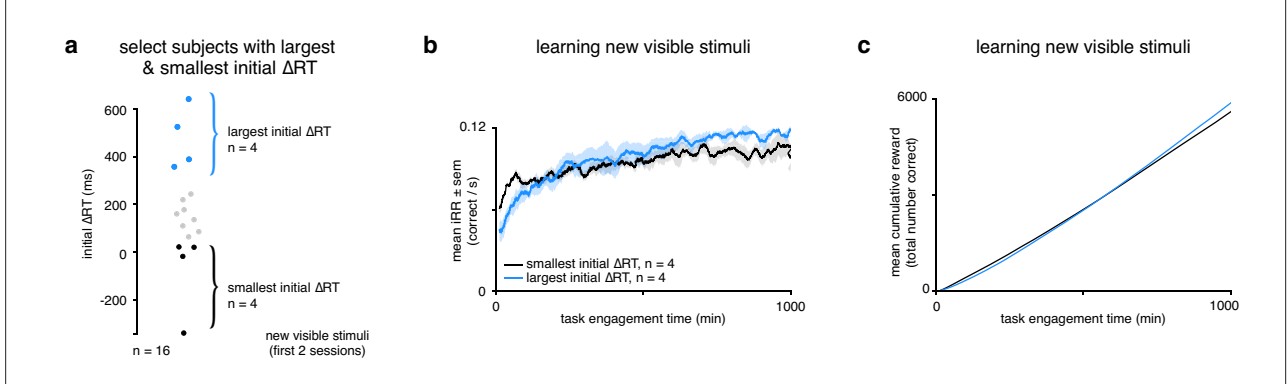

**Figure 7.** Rats that slowed down reaction times the most reached a higher instantaneous reward rate sooner and collected more reward. (a) Schematic showing segregation of top 25% of subjects ($n = 4$) with the largest initial $\Delta RTs$ for the new visible stimuli and the bottom 25% of subjects ($n = 4$) with the smallest initial $\Delta RTs$. Initial $\Delta RTs$ were calculated as an average of the first two sessions for all subjects. (**b**) Mean *iRR* for subjects with largest and smallest mean changes in reaction time across task engagement time. (**c**) Mean cumulative reward over task engagement time for subjects as in b.

tracking *iRR* and cumulative reward for the rats in the learnable stimuli setting with the largest (blue, $n = 4$) and smallest (black, $n = 4$) 'self-imposed' change in $RT$ (*Figure 7a*). The rats with the largest change started with a lower but ended with a higher mean iRR, and collected more cumulative reward (*Figure 7b and c*). Thus, in the voluntary setting there is a clear relationship between $RT$, learning speed, and its total reward benefits.

## Discussion

### Summary and limitations

Our theoretical and empirical results identify a trade-off between the need to learn rapidly and the need to accrue immediate reward in a perceptual decision making task. We find that rats adapt their decision strategy to improve learning speed and approximately maximize total reward, effectively navigating this trade-off over the total period of task engagement. In our experiments, rats responded slowly upon encountering novel stimuli, but only when there was a visual stimulus to learn from. This result indicates that they chose to respond more slowly in order to learn quickly, and only made the investment when learning was possible. This behavior requires foregoing both a cognitively easier strategy – fast random choice – and relinquishing a higher immediately available reward for several sessions spanning multiple days. By imposing different response times in groups of animals, we empirically verified our theoretical prediction that slow responses lead to faster learning and greater total reward in our task. These findings collectively show that rats exhibit cognitive control of the learning process, that is, the ability to engage in goal-directed behavior that would otherwise conflict with default or more immediately rewarding responses (*Dixon et al., 2012*, *Shenhav et al., 2013*; *Shenhav et al., 2017*, *Cohen et al., 1990*; *Cohen and Egner, 2017*).

Our high-throughput behavioral study with a controlled training protocol permits examination of the entire trajectory of learning, revealing hallmarks of non-greedy decision making. Nonetheless, it is accompanied by several experimental limitations. Our estimation of SNR improvements during learning relies on the DDM. Importantly, while this approach has been widely used in prior work (*Brunton et al., 2013*; *Ratcliff et al., 2006*; *Balci et al., 2011b*; *Drugowitsch et al., 2019*; *Petrov et al., 2011*), our conclusions are predicated on this model's approximate validity for our task. Future work could address this issue by using a paradigm in which learners with different response deadlines are tested at the same fixed response deadline, equalizing the impact of stimulus exposure at test. This model-free paradigm is not trivial in rodents, because response deadlines cannot be rapidly instructed. Our study also focuses on one visual perceptual task. Further work should verify our findings with other perceptual tasks across difficulties, modalities, and organisms.

To understand possible learning trajectories, we introduced a theoretical framework based on an RNN, and from this derived an LDDM. The LDDM extends the canonical drift-diffusion framework to incorporate long-term perceptual learning, and formalizes a trade-off between learning speed and instantaneous reward. However, it remains approximate and limited in several ways. The LDDM builds off the simplest form of a DDM, while various extensions and related models have been proposed to better fit behavioral data, including urgency signals (*Ditterich, 2006*; *Cisek et al., 2009*; *Deneve, 2012*; *Hanks et al., 2011*; *Drugowitsch et al., 2012*), history-dependent effects (*Busse et al., 2011*; *Scott et al., 2015*; *Akrami et al., 2018*; *Odoemene et al., 2018*; *Pinto et al., 2018*; *Lak et al., 2018*; *Mendonça et al., 2018*), imperfect sensory integration (*Brunton et al., 2013*), confidence (*Kepecs et al., 2008*; *Lak et al., 2014*; *Drugowitsch et al., 2019*), and multi-alternative choices (*Krajbich and Rangel, 2011*, *Tajima et al., 2019*). Prior work in the DDM framework has investigated learning dynamics with a Bayesian update and constant thresholds across trials (*Drugowitsch et al., 2019*). Our framework uses simpler error-corrective learning rules, and focuses on how the decision threshold policy over many trials influences long-term learning dynamics and total reward. Future work could combine these approaches to understand how Bayesian updating on each trial would change long-term learning dynamics, and potentially, the optimality of different threshold strategies.

More broadly, it remains unclear whether the drift-diffusion framework in fact underlies perceptual decision making, with a variety of other proposals providing differing accounts (*Gold and Shadlen, 2007*, *Zoltowski et al., 2019*; *Stine et al., 2020*). We speculate that the qualitative learning speed/instantaneous reward rate trade-off that we formally derive in the LDDM would also arise in other models of within-trial decision making dynamics. In addition, on a long timescale over many trials,

the LDDM improves performance through error-corrective learning. Future work could investigate learning dynamics under other proposed learning algorithms such as feedback alignment (*Lillicrap et al., 2016*), node perturbation (*Williams, 1992*), or reinforcement learning (*Law and Gold, 2009*). Additionally, the LDDM does not currently include a meta-learning component with which the agent can dynamically gauge the learnability of the task explicitly in order to set its decision threshold. Instead, the LDDM assumes a 'learnability prior' implemented as a high initial threshold condition for every new task. This limitation could be solved with a Bayesian observer that predicts learnability based on experience and controls the threshold accordingly. One potential avenue in this direction would be the implementation of the *learned value of control* theory, which provides a mechanism through which an agent can compare stimulus features to those it has encountered in the past in order to determine control allocation (*Lieder et al., 2018*). Moreover, the link between the LDDM and cognitive control is implicit: we interpret the choice of threshold in the DDM as a control process (a higher threshold than is optimal reflects control because it requires foregoing present reward in the service of future reward). Future modeling work should make the choice of control explicit, taking into account the inherent cost of control (*Shenhav et al., 2013*), and then using that choice to determine the decision threshold. Doing so would allow control to not only reflect the choice of threshold, as we have done, but also as a gain term on the drift rate (*Leng et al., 2021*), which may more completely capture control's role in two-choice decisions.

## Explore/exploit trade-off

Conceptually, the learning speed/instantaneous reward rate trade-off is related to the explore/exploit trade-off common in reinforcement learning, but differs in level of analysis. As traditionally framed in reinforcement learning, an agent has the option of maximizing reward based on its current information (exploitation), or of reaching a potentially larger future reward by expanding its current information (exploration). When framed this way, learning is an act of exploration. However, as framed in our study, learning is a systematic, directed strategy (or 'action'), that is, exploitation, employed in order to maximize total future reward. The reconciliation between these seemingly contradictory accounts occurs at the meta-level: when an agent is aware that learning is the optimal strategy to maximize total future discounted reward, it is exploiting a strategy that trades learning speed for instantaneous reward rate. However, when that agent is not yet aware whether it can learn, then it must explore this question (i.e. meta-learn) before deciding whether it should exploit an explicit learning strategy ('exploitation of exploration') that will also come at the cost of instantaneous reward. Although explained sequentially, these two mechanisms can occur in parallel (i.e. an agent constantly probing its learning prospects). One intriguing finding is that state-of-the-art deep reinforcement learning agents, which succeed in navigating the traditional explore/exploit dilemma on complicated tasks like Atari games (*Mnih et al., 2016*), nevertheless fail to learn perceptual decisions like those considered here (*Leibo et al., 2018*). This may be because exploration and exploitation can mean different things depending on the level of analysis, and efficiently learning a perceptual task may require the 'exploitation of exploration'. Our findings may thus offer routes for improving these artificial systems.

## Cognitive control

In order to navigate the learning speed/instantaneous reward rate trade-off, our findings suggest that rats deploy cognitive control of the learning process. Two main features of cognitive control govern its use: it is limited (*Shenhav et al., 2017*), and it is costly (*Krebs et al., 2010*; *Padmala and Pessoa, 2011*, *Kool et al., 2010*, *Dixon et al., 2012*; *Westbrook et al., 2013*; *Kool and Botvinick, 2018*; *Westbrook et al., 2019*). If control is costly, then its application needs to be justified by the benefits of its application. The *expected value of control* (EVC) theory posits that control is allocated in proportion to the EVC (*Shenhav et al., 2013*). Previous work demonstrated that rats are capable of the economic reasoning required for optimal control allocation (*Niyogi et al., 2014a*; *Niyogi et al., 2014b*; *Sweis et al., 2018*). We demonstrated that rats incur a substantial initial instantaneous reward rate opportunity cost to learn the task more quickly, foregoing a cognitively less demanding fast random strategy that would yield higher initial rewards. Rather than optimizing instantaneous reward rate, which has been the focus of prior theories (*Gold and Shadlen, 2002*, *Balci et al., 2011b*; *Bogacz et al., 2006*), our analysis suggests that rats approximately optimize total reward over task engagement. Relinquishing initial reward to learn faster, a cognitively costly strategy, is justified by a larger

total reward over task engagement. This pattern of behavior matches theoretical predictions of the value of learning based on a recent expansion of the EVC theory (*Masís et al., 2021*).

Assessing the expected value of learning in a new task requires knowing how much can be learned, how quickly one can learn, and for how long the task will be performed (*Masís et al., 2021*). None of these quantities is directly observable upon first encountering a new task, leading to the question of how rodents know to slow down in one task but not another. Importantly, rats only traded reward for information when learning was possible, a result in line with data demonstrating that humans are more likely to trade reward for information during long experimental time horizons, when learning is more likely (*Wilson et al., 2014*). Monkeys also reduce their reliance on expected value during decision making in order to explore strategically when it is deemed beneficial (*Jahn et al., 2022*). Moreover, previous work has highlighted the explicit opportunity cost of longer deliberation times (*Drugowitsch et al., 2012*), a trade-off that will differ during learning and at asymptotic performance, as we demonstrate here. One possibility is that rats estimate learnability and task duration through meta-learning processes that learn to estimate the value of learning through experience with many tasks (*Finn et al., 2017*; *Wang et al., 2018*; *Metcalfe, 2009*). The amount of control allocated to learning the current task could be proportional to its estimated value, determined based on similarity to previous learning situations and their reward outcomes and control costs (*Lieder et al., 2018*). Some of this bias for new information, termed curiosity, could be partly endogenous, serving as a useful heuristic for organisms outside of the lab, where rewards are sparse and action spaces are broad (*Gottlieb and Oudeyer, 2018*). Previous observations of suboptimal decision times in humans analogous to those we observed in rats might reflect incomplete learning, or subjects who think they still have more to learn (*Balci et al., 2011b*; *Bogacz et al., 2010*; *Cohen et al., 1990*). Future work could test further predictions emerging from a control-based theory of learning. An agent should assess both the predicted duration of task engagement and the predicted difficulty of learning in order to determine the optimal decision making strategy early in learning, and this can be tested by, for instance, manipulating the time horizon and difficulty of the task. From a control-based perspective, the expected reward from a task is also relevant to control allocation. Indeed, recent work in humans shows that externally motivating learners with the prospect of a test at the end of a task led to a much higher allocation of time on the harder-to-learn items compared to the case when learners were not warned of a test (*Ten et al., 2020*).

The trend of a decrease in response time and an increase in accuracy through practice – which we observed in our rats – has been widely observed for decades in the skill acquisition literature, and is known as the *Law of Practice* (*Thorndike, 1913*, *Newell and Rosenbloom, 1981*, *Logan, 1992*, *Heathcote et al., 2000*). Accounts of the Law of Practice have posited a cognitive control-mediated transition from shared/controlled to separate/automatic representations of skills with practice (*Posner and Snyder, 1975*, *Shiffrin and Schneider, 1977*, *Cohen et al., 1990*). On this view, control mechanisms are a limited, slow resource that impose unwanted processing delays. Our results suggest an alternative non-mutually exclusive reward-based account for why we may so ubiquitously observe the Law of Practice. Slow responses early in learning may be the goal of cognitive control, as they allow for faster learning, and faster learning leads to higher total reward. When faced with the ever-changing tasks furnished by naturalistic environments, it is the speed of learning which may exert the strongest impact on total reward.

## Bounded optimality

More broadly, the optimization of behavior, not in a vacuum, but in the context of one's constraints – intrinsic and environmentally determined – underlies several general theories of cognition, including theories that explain the allocation of cognitive control (*Shenhav et al., 2013*; *Lieder et al., 2018*), the selection of decision heuristics (*Gigerenzer, 2008*), and the rationale of seemingly irrational economic choices (*Kahneman and Tversky, 1979*, *Juechems et al., 2021*). These theories are instances of bounded optimality – a prominent theoretical framework of biological and artificial cognition stating that an agent is optimal when it maximizes reward per unit time within the limitations of its computational architecture (*Russell and Subramanian, 1994*, *Lewis et al., 2014*; *Gershman et al., 2015*; *Griffiths et al., 2015*; *Bhui et al., 2021*; *Summerfield and Parpart, 2021*).

Instances of this framework typically assume that cognitive constraints remain fixed and, more so, that agents do not take alterations of these constraints into account when choosing what to do. There

exists, however, a novel theoretical avenue within this framework. An agent can optimize its behavior not only through maximization of reward *within* constraints, but also through the *minimization of those constraints themselves*. If an agent can change itself to minimize its constraints by, for example, improving its perceptual representations through learning, the future reward prospects of doing so should be considered in its current choices, even if it is at the expense of current reward. Intelligent agents, like humans, can and do change themselves through learning in order to improve future reward prospects. Our study formalizes this phenomenon in the context of two-choice perceptual decisions, but much work remains to be done in other contexts, modalities, and organisms.

## Methods
### Behavioral training
#### Subjects

All care and experimental manipulation of animals were reviewed and approved by the Harvard Institutional Animal Care and Use Committee (IACUC), protocol 27–22. We trained animals on a high-throughput visual object recognition task that has been previously described (*Zoccolan et al., 2009*). A total of 44 female Long-Evans rats were used for this study, with 38 included in analyses. Twenty-eight rats (AK1–12 and AL1–16) initiated training on stimulus pair 1, and 26 completed it (AK8 and AL12 failed to learn). Another 8 animals (AM1–8) were trained on stimulus pair 1 but were not included in the initial analysis focusing on asymptotic performance and learning (*Figure 1d and e*; *Figure 2*) because they were trained after the analyses had been completed. Subjects AM5–8, although trained, did not participate in other behavioral experiments so do not appear in this study. Sixteen animals (AL1–8, AL13–16, and AM1–8) participated in learning stimulus pair 2 ('new visible stimuli'; canonical-only training regime) while 10 animals (AK1–3, 5–7, 9–12) initially participated in viewing transparent (alpha = 0; AK1, 3, 6, 7, 11) or near-transparent stimuli (alpha = 0.1; AK2, 5, 9, 10, 12), with the subjects sorted randomly into each group. The transparent and near-transparent groups were aggregated but two animals from the near-transparent group were excluded for performing above chance (AK5 and AK12) as this experiment focused on the effects of stimuli that could not be learned. The same 16 animals used for stimulus pair 2 were used for learning stimulus pair 3 under two different reaction time restrictions in which the subjects were sorted randomly. One rat (AL1) was excluded from the outset for not having learned stimulus pair 2. Two additional rats (AL4 and AL7) were excluded for not completing enough trials during practice sessions with the new reaction time restrictions. A final rat (AM1) was excluded because she failed to learn the task. The 12 remaining rats were grouped into seven subjects required to respond above (AL3, AL8, AL13, AL15, AL16, AM3, AM4) and five subjects required to respond below their individual average reaction times (AL2, AL5, AL6, AL14, AM2). Finally, eight rats (AN1–8) were trained on a simplified training regime ('canonical only') used as a control for the typical 'size and rotation' training object recognition regime (described below). *Table 1* summarizes individual subject participation across behavioral experiments.

#### Behavioral training boxes

Rats were trained in high-throughput behavioral training rigs, each made up of four vertically stacked behavioral training boxes. In order to enter the behavioral training boxes, the animals were first individually transferred from their home cages to temporary plastic housing cages that would slip into the behavioral training boxes and snap into place. Each plastic cage had a porthole in front where the animals could stick out their head. In front of the animal in the behavior boxes were three easily accessible stainless steel lickports electrically coupled to capacitive sensors, and a computer monitor (Dell P190S, Round Rock, TX, USA; Samsung 943-BT, Seoul, South Korea) at approximately 40° visual angle from the rats' location. The three sensors were arranged in a straight horizontal line approximately a centimeter apart and at mouth-height for the rats. The two side ports (L/R) were connected to syringe pumps (New Era Pump Systems, Inc NE-500, Farmingdale, NY, USA) that would automatically dispense water upon a correct trial. The center port was connected to a syringe that was used to manually dispense water during the initial phases of training (see below). Each behavior box was equipped with a computer (Apple Macmini 6,1 running OsX 10.9.5 [13F34] or Macmini 7.1 running OSX El Capitan 10.11.13, Cupertino, CA, USA) running MWorks, an open source software for running real-time behavioral experiments (MWorks 0.5.dev [d7c9069] or 0.6 [c186e7], The MWorks Project

**Table 1.** Individual animal participation across behavioral experiments.

| Animal | Sex | Stimulus pair 1 | Stimulus pair 2 | Transparent stimuli | Stimulus pair 3 |
|--------|-----|-----------------|-----------------|---------------------|-----------------|
| AK1 | F | Size and rotation | Alpha = 0 | | |
| AK2 | F | Size and rotation | Alpha = 0.1 | | |
| AK3 | F | Size and rotation | Alpha = 0.0 | | |
| AK4 | F | Size and rotation | | | |
| AK5 | F | Size and rotation | Alpha = 0.1 (excluded)[‡] | | |
| AK6 | F | Size and rotation | Alpha = 0 | | |
| AK7 | F | Size and rotation | Alpha = 0 | | |
| AK8 | F | Size and rotation (excluded)* | | | |
| AK9 | F | Size and rotation | Alpha = 0.1 | | |
| AK10 | F | Size and rotation | Alpha = 0.1 | | |
| AK11 | F | Size and rotation | Alpha = 0.0 | | |
| AK12 | F | Size and rotation | Alpha = 0.1 (excluded)[‡] | | |
| AL1 | F | Size and rotation | Canonical only | (Excluded)[§] | |
| AL2 | F | Size and rotation | Canonical only | Below | |
| AL3 | F | Size and rotation | Canonical only | Above | |
| AL4 | F | Size and rotation | Canonical only | Below (excluded) [¶] | |
| AL5 | F | Size and rotation | Canonical only | Below | |
| AL6 | F | Size and rotation | Canonical only | Below | |
| AL7 | F | Size and rotation | Canonical only | Below (excluded)[¶] | |
| AL8 | F | Size and rotation | Canonical only | Above | |
| AL9 | F | Size and rotation | | | |
| AL10 | F | Size and rotation | | | |
| AL11 | F | Size and rotation | | | |
| AL12 | F | Size and rotation (excluded)* | | | |
| AL13 | F | Size and rotation | Canonical only | Above | |
| AL14 | F | Size and rotation | Canonical only | Below | |
| AL15 | F | Size and rotation | Canonical only | Above | |
| AL16 | F | Size and rotation | Canonical only | Above | |
| AM1 | F | Size and rotation[†] | Canonical only | Below (excluded)** | |
| AM2 | F | Size and rotation[†] | Canonical only | Below | |
| AM3 | F | Size and rotation[†] | Canonical only | Above | |
| AM4 | F | Size and rotation[†] | Canonical only | Above | |
| AM5 | F | Size and rotation[†] | | | |
| AM6 | F | Size and rotation[†] | | | |
| AM7 | F | Size and rotation[†] | | | |
| AM8 | F | Size and rotation[†] | | | |
| AN1 | F | Canonical only | | | |
| AN2 | F | Canonical ony | | | |
| AN3 | F | Canonical only | | | |

*Table 1 continued on next page*

*Table 1 continued*

| Animal | Sex | Stimulus pair 1 | Stimulus pair 2 | Transparent stimuli | Stimulus pair 3 |
|--------|-----|-----------------|-----------------|---------------------|-----------------|
| AN4 | F | Canonical only | | | |
| AN5 | F | Canonical only | | | |
| AN6 | F | Canonical only | | | |
| AN7 | F | Canonical only | | | |
| AN8 | F | Canonical only | | | |

*Failed to learn task.

†Not included in initial learning experiment.

‡Above chance for near-transparent stimuli.

§Failed to learn previous stimuli.

¶Not enough practice trials with reaction time restrictions.

**Failed to learn stimuli with reaction time restrictions.

https://mworks.github.io/). The capacitive sensors (Phidget Touch Sensor P/N 1129_1, Calgary, Alberta, Canada) were controlled by a microcontroller (Phidget Interface Kit 8/8/8P/N 1018_2) that was connected via USB to the computer. The syringe pumps were connected to the computer via an RS232 adapter (Startech RS-232/422/485 Serial over IP Ethernet Device Server, Lockbourne, OH, USA). To allow the experimenter visual access to the rats' behavior, each box was, in addition, illuminated with red LEDs, not visible to the rats.

## Habituation

Long-Evans rats (Charles River Laboratories, Wilmington, MA, USA) of about 250 g were allowed to acclimate to the laboratory environment upon arrival for about a week. After acclimation, they were habituated to humans for 1 or 2 days. The habituation procedure involved petting and transfer of the rats from their cage to the experimenter's lap until the animals were comfortable with the handling. Once habituated to handling, the rats were introduced to the training environment. To allow the animals to get used to the training plastic cages, the feedback sounds generated by the behavior rigs, and to become comfortable in the behavior training room, they were transferred to the temporary plastic cages used in our high-throughput behavioral training rigs and kept in the training room for the duration of a training session undergone by a set of trained animals. This procedure was repeated after water deprivation, and during the training session undergone by the trained animals, the new animals were taught to poke their head out of a porthole available in each plastic cage to receive a water reward from a handheld syringe connected to a lickport identical to the ones in the behavior training boxes in the training rigs. Once the animals reliably stuck their head out of the porthole (1 or 2 days) and accessed water from the syringe, they were moved into the behavior boxes.

## Early shaping

On their first day in the behavior boxes, rats were individually tutored as follows: Water reward was manually dispensed from the center lickport which is normally used to initiate a trial. When the animal licked the center lickport, a trial began. After a 500 ms tone period, one of two visual objects (stimulus pair 1) appeared on the screen (large front view, degree of visual angle 40°) chosen pseudo-randomly (three randomly consecutive presentations of one stimulus resulted in a subsequent presentation of the other stimulus). This appearance was followed by a 350 ms minimum reaction time that was instituted to promote visual processing of the stimuli. If the animal licked one of the side (L/R) lickports during this time, then the trial was aborted, there would be a minimum intertrial time (1300 ms), and the process would begin again.

At the time of stimulus presentation, a free water reward was dispensed from the correct side (L/R) lickport. If the animals licked the correct side lickport within the allotted amount of time (3500 ms) then an additional reward was automatically dispensed from that port. This portion of training was meant to begin teaching the animals the task mechanics, that is to first lick the center port, and then one of the two side ports.

After the rats were sufficiently engaged with the lickports and began self-initiating trials by licking the center lickport (usually 1 to several days, determined by experimenter) no more water was dispensed manually through the center lickport, but the free water rewards from the side lickports were still given. Once the rats were self-initiating enough trials without manual rewards from the center lickport (>200 per session), the free reward condition was stopped, and only correct responses were rewarded.

## Training

Data collection for this study began once the rats had demonstrated proficiency of the task mechanics (as described above). The training curriculum followed was similar to that by *Zoccolan et al., 2009*. Rats performed the task for about 2 hr daily. Initially, the rats were only presented with large front views (40° visual angle, 0° of rotation) of the two stimuli (stimulus pair 1). Once the rats reached a performance level of ≥70% with these views, the stimuli decreased in size to 15° visual angle in a stair-cased fashion with steps of 2.5° visual angle. Once the rats reached 15° visual angle, rotations of the stimuli to the left or right were staircased in steps of 5° at a constant size of 30° visual angle. Once the rats reached ±60° of rotation, they were considered to have completed training and were presented with random transformations of the stimuli at different sizes (15°–40° visual angle, step = 15°; 0° of rotation) or different rotations (-60° to +60° of rotation, step = 15°; 30° visual angle). After this point, 10 additional training sessions were collected to allow the animals' performance to stabilize with this expanded stimulus set.

During training, there was a bias correction that tracked the animals' tendency to be biased to one side. If biased, stimuli mapped to the unbiased side were presented for a maximum of three consecutive trials. For example, if the bias correction detected an animal was biased to the right, the left-mapped stimulus would appear three trials at a time in a non-random fashion and the animals' performance would drop from 50% to 25%, reducing the advantageousness of a biased strategy dramatically. If the animals continued to exhibit bias after one or two sessions of bias correction, then the limit was pushed to five consecutive trials. Once the bias disappeared, stimulus presentation resumed in a pseudo-random fashion.

The left/right mapping of the stimuli to lickports was counterbalanced across animals, ruling out any effects related left/right stimulus-independent biases, or left/right-independent stimulus bias across animals.

## Training regime comparison

Although object recognition is supposed to be a fairly automatic process (*Cox, 2014*), it is possible that the 14 possible presentations of each stimulus of stimulus pair 1 (6 sizes at constant rotation, and 8 rotations at constant size) varied in difficulty. To rule out any possible difficulty effects during training and at asymptotic performance, We trained $n = 8$ different rats to asymptotic performance on the task but only on large, front views of the visual objects (*Figure 2—figure supplement 1a*). We compared the learning and asymptotic performance of the 'size and rotation' cohort and the 'canonical only' cohort across a wide range of behavioral measures. During learning, animals in both regimes followed similar learning trajectories in speed-accuracy space (*Figure 2—figure supplement 1b*), and clustered around the OPC at asymptotic performance (*Figure 2—figure supplement 1c*). Comparisons of accuracy, reaction time, and fraction maximum instantaneous reward rate trajectories during learning and at averages asymptotic performance revealed no detectable differences (*Figure 2—figure supplement 1d—f*). Total trials per session, and voluntary intertrial intervals after error trials did show slightly varied trajectories during learning, though there were no differences in their means after learning (*Figure 2—figure supplement 1g, h*). The difference in total trials per session could be unrelated to the difference in training regimes. The difference in voluntary intertrial intervals, however, could be related to the introduction of different sizes and rotations: a sudden spike in this metric is seen about halfway through normalized sessions and decays over time. If this is the case, it is a curious result that rats choose to display their purported 'surprise' in between trials, and not during trials, as we found no difference in the reaction time trajectories. Both training regimes had overlapping fraction trials ignored metrics during learning, with a sharp decrease after the start, and a small significant difference in their number at asymptotic performance (*Figure 2—figure supplement 1i*). We point out the fact that we do not consider voluntary intertrial intervals nor ignored trials in our analysis, so

the differences between the regimes do not affect our conclusions. Overall, these results suggest that there is not a measurable or relevant difficulty effect based on our training regime with a variety of stimulus presentations.

## Stimulus learnability experiment

Transparent stimuli. In order to assess how animals behaved in a scenario with non-existent learning potential, a subset of already well-trained animals were presented with transparent ($n = 5$, alpha = 0) or near-transparent ($n = 5$, alpha = 0.1) versions of the familiar stimulus pair 1 for a duration of 11 sessions. Before these sessions, 4 sessions with stimulus pair 1 at full opacity (alpha = 1) were conducted to ensure animals could perform the task adequately before the manipulation. We predicted that the near-transparent condition would segregate animals into two groups, those that could perform the task and those that could not, based on each individual's perceptual ability. The animals in the near-transparent condition that remained around chance performance ($n = 3$, rat AK2, AK9, and AK10) were grouped with the animals from the transparent condition, while those that performed well above chance ($n = 2$, rat AK5 and AK12) were excluded.

Reaction times were predicted to decrease during the course of the experiment, so to measure the change most effectively, the minimum reaction time requirement of 350 ms was removed. However, removing the requirement could lead to reduced reaction times regardless of the presented stimuli. To be able to measure whether the transparent stimuli led to a significant difference in reaction times compared to visible stimuli, we ran sessions with visible stimuli with no reaction time requirement for the same animals and compared these reaction times with those from the transparent condition. We found that the aggregate reaction time distributions were significantly different (*Figure 6—figure supplement 1a*). A comparison of vincentized reaction times revealed that there was a significant difference in the fastest reaction time decile (*Figure 6—figure supplement 1b*), confirming that reaction times decreased significantly during presentation of transparent stimuli.

New visible stimuli. In order to assess how animals behaved in a scenario with high learning potential, a subset ($n = 16$) of already well-trained animals on stimulus pair 1 were presented with a never before seen stimulus pair (stimulus pair 2) for a duration of 13 sessions. Before these sessions, 5 sessions with the familiar stimulus pair 1 were recorded immediately preceding the stimulus pair 2 sessions in order to compare performance and reaction time after the manipulation for every animal. Previous pilot experiments showed that the animals immediately assigned a left/right mapping to the new stimuli based on presumed similarity to previously trained stimulus pair, so in order to enforce learning, the left/right mapping contrary to that predicted by the animals in the pilot tests was chosen. Because of this, animals typically began with an accuracy below 50%, as they first had to undergo reversal learning for their initial mapping assumptions. Because the goal of this experiment was to measure effects during learning and not demonstrate invariant object recognition, the new stimuli were presented in large front views only (visual angle = 40°, rotation = 0°).

## Behavioral data analysis

### Software

Behavioral psychophysical data was recorded using the open-source MWorks 0.5.1 and 0.6 software (https://mworks.github.io/downloads/). The data were analyzed using Python 2.7 with the pymworks extension. We employed the hierarchical estimation of the DDM in python (HDDM) package for DDM fits (*Wiecki et al., 2013*). To measure stimulus-independent psychophysical strategies such as bias and perseverance, we employed PsyTrack, a generalized linear model package for fitting dynamic psychophysical models to behavioral data (*Roy et al., 2021*) in conjunction with Python 3.8.

### DDM fit

In order to verify that our behavioral data could be modeled as a drift-diffusion process, the data were fit with an HDDM (*Wiecki et al., 2013*), permitting subsequent analysis (such as comparison to the OPC) based on the assumption of a drift-diffusion process. To verify that a DDM was appropriate for our data, we fit a simple DDM to 10 asymptotic sessions after learning stimulus pair 1 for $n = 26$ subjects (*Figure 1—figure supplement 3*). In order to assess parameter changes across learning, we fit DDMs to the stimulus pair 1 experiment and the stimulus pair 2 experiment where the learning epochs were treated as conditions in each experiment. This allowed us to hold some parameters

constant while conditioning others on learning. We fit both simple DDMs and DDMs with drift rate variability to the two experiments, allowing drift rate, threshold, and drift rate variability to vary with learning epoch. In particular, we fit three broad types of models: (1) simple DDMs (*Figure 4—figure supplement 2*), (2) DDMs + fixed drift rate variability (*Figure 4—figure supplement 3*), and (3) DDMs + drift rate variability that varied freely with learning epoch (*Figure 4—figure supplement 4*). For each of the types of models we held drift constant, threshold constant, or allowed both to vary with learning. The best fits, as determined by the deviance information criterion (DIC), came from models where we allowed both drift and threshold to vary with learning; the addition of drift rate variability did not appear to improve model fits (*Figure 4—figure supplement 1*). For both learning experiments, drift rates increased and thresholds decreased by the end of learning, in agreement with previous findings (*Ditterich, 2006*; *Ratcliff et al., 2006*; *Dutilh et al., 2009*; *Balci et al., 2011b*; *Liu and Watanabe, 2012*; *Zhang and Rowe, 2014*). In addition, for the transparent stimuli experiment we fit a DDM that allowed drift rate, threshold, drift rate variability, and $T_0$ to vary with learning phase in order to observe the changes in drift rate and threshold (*Figure 6—figure supplement 3*).

## PsyTrack fit

In order to estimate stimulus-independent psychophysical strategies in the transparent stimulus experiment (*Figure 6—figure supplement 4*), we used PsyTrack to fit a generalized linear model to our behavioral data (*Roy et al., 2021*). The model assigns weights to user-determined input variables to explain the output variable. The output variable consists of a vector for left/right choices on every trial for an individual subject, where left = 0 and right = 1 (or 1 and 2). PsyTrack automatically calculates weights on bias, with a positive weight indicating a rightward bias, and a negative weight indicating a leftward bias. We fit the model by providing it with three explanatory input variables: stimulus, perseverance, and win-stay/lose-switch. For the input variables, the left/right coding differed from that for the output variable as per the model documentation (left = -1, right = +1). The stimulus variable indicated the stimulus that appeared on that trial (stimulus A = -1, stimulus B = +1). However, the left/right mapping of these stimuli was counterbalanced across subjects, so depending on the mapping, subjects could have a strong positive or negative weight, both indicating the stimuli explained choices. The perseverance variable indicated the left/right location of the subject's choice on the previous trial. The win-stay/lose-switch variable indicated the location of the correct choice on the previous trial. For both perseverance and win-stay/lose-switch, positive weights indicate the predicted presence of perseverance and win-stay/lose-switch rather than left/right information.

## Behavioral metrics

Error rate (*ER*) was calculated by dividing the number of error trials by the number of total trials (error + correct) within a given window of trials in a full behavioral training session. Accuracy was calculated as $1 - ER$.

$$ER = \frac{\text{error trials}}{\text{total trials}} \tag{13}$$

$$\text{accuracy} = 1 - ER \tag{14}$$

Reaction time (*RT*) for one trial was measured by subtracting the time of the first lick on a response lickport from the stimulus onset time on the computer monitor. Mean *RT* was calculated by averaging reaction times across trials within a given window of trials or the trials in a full behavioral training session.

$$RT = \frac{1}{n} \sum_{\text{trial } i=1}^{\text{trial } n} RT_i \tag{15}$$

Vincentized reaction time is one method to report aggregate reaction time data meant to preserve individual distribution shape and be less sensitive to outliers in the group distribution (*Ratcliff, 1979*, *Blokland, 1998*), although some scientists have argued parametric fitting (with an ex-Gaussian distribution, for example) and parameter averaging across subjects outperforms Vincentizing as sample size increases (*Rouder and Speckman, 2004*; *Whelan, 2008*). Each subject's reaction time distribution is

divided into quantiles (e.g. deciles; similar to percentile, but between 0 and 1), and then the quantiles across subjects are averaged.

Decision time ($DT$) for one trial was measured by subtracting the non-decision time $T_0$ (see Estimating $T_0$) from $RT$. Mean $DT$ was calculated by subtracting $T_0$ from the mean $RT$ across trials within a given window of trials or the trials in a full behavioral training session.

$$DT = RT - T_0. \tag{16}$$

Post-error and correct non-decision task engagement times ($D_{err}$, $D_{corr}$) are defined (for notational simplicity) as the sum of non-decision time $T_0$ and the experimentally determined response-to-stimulus times after error $\tilde{D}_{err}$ and correct trials $\tilde{D}_{corr}$. Please see Determining $\tilde{D}_{err}$ and $\tilde{D}_{corr}$ for how we determined these experimental variables.

$$D_{err} = \tilde{D}_{err} + T_0 \tag{17}$$

$$D_{corr} = \tilde{D}_{corr} + T_0. \tag{18}$$

Mean normalized decision time ($DT/D_{err}$) was measured by dividing mean $DT$ by $D_{err}$, the sum of the non-decision time $T_0$ and $\tilde{D}_{err}$, the mean non-decision task engagement time in an error trial (see $D_{err}$, $D_{corr}$).

Mean difference in mean reaction time ($\Delta RT$) was calculated by subtracting the mean reaction time of a number of baseline sessions from the mean reaction time of an experimental session. A positive difference indicates an increase over baseline mean reaction time. The mean of the two immediately preceding sessions with stimulus pair 1 were subtracted from the mean reaction time of every session with stimulus pair 2 or transparent stimuli for every animal individually (**Figure 6d and e**). These differences were then averaged to get a mean difference in mean reaction time $\Delta RT$.

Mean instantaneous reward rate ($iRR$) (regularly referred to as just reward rate, $RR$) is defined as mean accuracy per mean time per trial (**Gold and Shadlen, 2002**):

$$iRR = \frac{\text{mean accuracy}}{\text{mean time per trial}}. \tag{19}$$

We define the average non-decision task engagement time per trial,

$$D_{tot} = (1 - ER)D_{corr} + ER \cdot D_{err}. \tag{20}$$

The mean instantaneous reward rate is then (see Equation A26 in **Bogacz et al., 2006**)

$$iRR = \frac{1 - ER}{DT + D_{tot}}. \tag{21}$$

N.B. Because our study hinges on the important difference between present, future and cumulative rewards and tracks changes in $ER$ and $DT$ over learning, we write what is traditionally referred to as reward rate $RR$ as instantaneous reward rate $iRR$ to emphasize these differences. Because $ER$ and $DT$ can change throughout learning, reward rate as traditionally defined only captures an 'instant' in a learning trajectory.

Mean total correct trials is a model-free measure of the reward attained by the animals within a given window of trials. Every correct response yields an identical water reward, hence, reward can be counted by counting correct responses across trials. For one subject $a \in [1, 2, 3, \ldots, k]$, total correct trials at trial $n$ are the sum of correct trials up to trial $n$:

$$c_n^a = \sum_{\text{trial } i=1}^{\text{trial } n} o_i^a, \tag{22}$$

where $o_i^a$ is an element in a vector $o^a$ containing the outcomes of those trials $o^a = [o_1^a, o_2^a, o_3^a, \ldots, o_n^a]$. For correct and error responses $o_n^a = 1$ and 0 respectively (e.g. $o_n^a = [0, 0, 1, 1, 0, \ldots, 1]$).

Mean total correct trials up to trial $n$ is calculated by taking the average of total correct trials across all animals $k$ up to trial $n$.

$$\langle c_n \rangle = \frac{1}{k} \sum_{\text{trial } i=1}^{\text{trial } n} c_i^1 + c_i^2 + c_i^3 + \ldots + c_i^k. \tag{23}$$

Mean cumulative reward is a measure of the reward attained by the animals within a given window of trials. To calculate this quantity, a moving average of $RT$ and accuracy for a given window size are first calculated for every animal individually. To avoid averaging artifacts, only values a full window length from the beginning are considered. Given these moving averages, $iRR$ is then calculated for every animal and subsequently averaged across animals to get a moving average of mean reward rate. To calculate the mean cumulative reward, a numerical integral over a particular task time, such as *task engagement time* (see Measuring task time), is then calculated using the composite trapezoidal rule.

SNR is a measure of an agent's perceptual ability in a discrimination task. Given an animal's particular $ER$ and $DT$, we use an equation to infer its SNR $\bar{A}$ deduced from standard DDM equations to infer its SNR (Equation 56 in *Bogacz et al., 2006*):

$$\bar{A}_{\text{infer}} = \frac{1 - 2ER}{2DT} \log \frac{1 - ER}{ER}.$$ (24)

The SNR equation defines a U-shaped curve that increases as *ERs* move away from 0.5. For cases early in learning where *ERs* were below 0.5 because of potential initial biases, we assumed the inferred SNR was negative (meaning the animals had to unlearn the biases in order to learn, and thus had a monotonically increasing SNR during learning).

SNR performance frontier is a measure of an agent's possible error rate and reaction time combinations based on their current perceptual ability. Because of the SAT, not all combinations of $ER$ and $DT$ are possible. Instead, performance is bounded by an agent's SNR $\bar{A}$ at any point in time, and their particular ($ER$, $DT$) combination will depend on their choice of threshold.

Given a fixed $D_{err}$ (as in the case of our experiment), this bound exists in the form of a performance frontier – the combination of all resultant *ERs* and mean normalized *DTs* possible given a fixed SNR $\bar{A}$ and all possible thresholds $\bar{z}$.

We can use $\bar{A}_{\text{infer}}$ (*Equation 24*) to calculate its performance frontier for a range of thresholds $\bar{z}$ [0, ∞) with standard equations from the DDM:

$$ER_{\bar{A}_{\text{infer}}} = \frac{1}{1 + e^{2\bar{z}\bar{A}_{\text{infer}}}}$$ (25)

$$DT_{\bar{A}_{\text{infer}}} = \bar{z} \tanh\left(\bar{z}\bar{A}_{\text{infer}}\right).$$ (26)

For every performance frontier there will be one unique ($ER_{\bar{A}_{\text{infer}}}$, $DT_{\bar{A}_{\text{infer}}}$) combination for which reward rate will be greatest, and it will lie on the OPC.

Fraction maximum instantaneous reward rate is a measure of distance to the OPC, that is, optimal performance. Given an animal's $ER$ and $DT$, we inferred their SNR and calculated their performance frontier as described above. We then divided the animal's reward rate by the maximum reward rate on their performance frontier, corresponding to the point on the OPC they could have attained given their inferred SNR $\bar{A}_{\text{infer}}$:

$$\text{fraction max } iRR = \frac{iRR_{ER, DT}}{\max iRR_{\bar{A}_{\text{infer}}}}.$$ (27)

Maximum instantaneous reward rate opportunity cost, like fraction maximum instantaneous reward rate, is also measure of distance to the OPC, that is, optimal performance, but it emphasizes the reward rate fraction given up by the subject given its current $ER$ and $DT$ combination along its SNR performance frontier. It is simply:

$$\text{max } iRR \text{ opportunity cost} = 1 - \text{fraction max } iRR.$$ (28)

Mean post-error slowing is a metric to account for the potential policy of learning by slowing down after error trials. In order to quantify the amount of post-error slowing in a particular subject, the subject's reaction times in a session are segregated into correct trials following an error, and correct trials following a correct choice, and separately averaged. The difference between these indicates the degree of post-error slowing present in that subject during that session.

$$\text{post-error slowing } (PES) = \langle RT_{\text{post-error correct trials}} \rangle - \langle RT_{\text{post-correct correct trials}} \rangle.$$ (29)

The mean post-error slowing for one session is thus the mean of this quantity across all subjects $k$.

$$\langle PES \rangle = \frac{PES^1 + PES^2 + PES^3 + ... + PES^k}{k}.$$ (30)

Left/right bias measures the extent to which a subject is biased to the left or right lickport regardless of the stimulus presented. For every individual, left or right choices for every trial are coded as a binary vector (left = 0, right = 1). The correct response side is also coded as a binary vector. Bias is calculated by taking the difference of these vectors. A Gaussian filter is then applied to smooth the bias vector over time, with negative numbers reflecting a left bias and positive numbers reflecting a right bias.

## Computing error

Within-subject session errors (e.g. *Figure 1d*) for accuracy and reaction times were calculated by bootstrapping trial outcomes and reaction times for each session. We calculated a bootstrapped standard error of the mean by taking the standard deviation of the distribution of means from the bootstrapped samples. A 95% confidence interval can be calculated from the distribution of means as well.

Across-subject session errors (e.g. *Figure 6d*) were computed by calculating the standard error of the mean of individual animal session means.

Across-subject sliding window errors (e.g. *Figure 6b*; *Figure 7b*) were calculated by averaging trials over a sliding window (e.g. 200 trials) for each animal first, then taking the standard error of the mean of each step across animals. Alternatively, the average could be taken across a quantile (e.g. first decile, second decile, etc.), and then the standard error of the mean of each quantile across animals was computed.

## Measuring task time

Trials are the smallest unit of behavioral measure in the task and are defined by one stimulus presentation accompanied by one outcome (correct, error) and one reaction time.

Sessions are composed of as many trials as an animal chooses to complete within a set window of wall clock time, typically around 2 hr once daily. An error rate (fraction of error trials over total trials for the session) and a mean reaction time can be calculated for a session.

Normalized sessions are a group of sessions (e.g. 1, 2, 3, …, 10) where a particular session's normalized index corresponds to its index divided by the total number of sessions in the group (e.g. 0.1, 0.2, 0.3, …, 1.0). Because animals may take different numbers of sessions to learn to criterion, a normalized index for sessions allows better comparison of psychophysical measurements throughout learning.

Stimulus viewing time measures the time that the animals are viewing the stimulus, defined as the sum of all reaction times up to trial $n$ as:

$$\text{stimulus viewing time} = \sum_{\text{trial } i=1}^{\text{trial } n} RT_i$$ (31)

$$\text{task engagement time} = \sum_{\text{trial } i=1}^{\text{trial } n} RT_i + n_{corr}\tilde{D}_{corr} + n_{err}\tilde{D}_{err}.$$ (32)

The sum of reaction times up to trial $n$ plus the sum of $\tilde{D}_{err}$ = 3136 ms and $\tilde{D}_{corr}$ = 6370 ms, the mandatory post-error and post-correct response-to-stimulus intervals, proportional to the number of error and correct trials ($n = n_{corr} + n_{nerr}$).

## Statistical analyses

*Figure 1d*: We wished to test whether the mean fraction maximum reward rate of our subjects over the 10 sessions after having completed training were significantly different from optimal performance. A Shapiro-Wilk test failed to reject ($p < 0.05$) a null hypothesis for normality for 18/26 subjects, with the following p-values (from left to right): (0.8162, 0.1580, 0.3746, 0.6985, 0.0025, 0.0467, 0.0040, 0.6522, 0.0109, 0.1625, 1.8178e-05, 0.0901, 0.7606, 0.0295, 0.0009, 0.2483, 0.5627, 0.0050, 0.4464, 0.6839, 0.5953, 0.0140, 0.1820, 0.1747, 0.6385, 0.2304). Thus, we conducted a one-sided Wilcoxon signed-rank test on our sample against 0.99, testing for the evidence that each subject's mean fraction

max reward rate was greater than 99% of the maximum (p<0.05), and obtained the following p-values (from left to right): (0.0025, 0.0025, 0.0025, 0.1013, 0.2223, 0.0063, 0.0047, 0.0025, 0.0025, 0.0025, 0.0025, 0.0571, 0.6768, 0.0047, 0.7125, 0.0372, 0.8794, 0.4797, 0.7125, 0.8987, 0.0372, 0.0109, 0.9975, 0.9766, 0.9917, 0.9975).

*Figure 5b*: We wished to test the difference in mean $RT$ between two randomly chosen groups of animals before and after an $RT$ restriction to assess the effectiveness of the restriction. A Shapiro-Wilk test did not support an assumption of normality for the 'below' group in either condition resulting in the following (W statistic, p-value) for the pre-$RT$ restriction 'above' and 'below' groups and post-RT restriction 'above' and 'below' groups: (0.9073, 0.3777), (0.6806, 0.0059), (0.8976, 0.3168), (0.6583, 0.0033). Hence, we conducted a Wilcoxon rank-sum test for the pre- and post-$RT$ restriction groups and found the pre-$RT$ restriction group was not significant (p=0.570) while the post-$RT$ restriction group was (p=0.007), indicating the two groups were not significantly different before the RT restriction, but became significantly different after the restriction.

*Figure 5d*: We wished to test the difference in accuracy between the 'above' and 'below' groups for every session of stimulus pair 3. A Shapiro-Wilk test failed to reject the assumption of normality (p<0.05) for any session from either condition (except session 4, 'above', which could be expected given there were 16 tests), with the following (W statistic, p-value) for [session: 'above', 'below'] by session: [1: (0.9340, 0.6240),(0.8959, 0.3068)], [2: (0.9381, 0.6522), (0.8460, 0.1130)], [3: (0.9631, 0.8291), (0.9058, 0.3676)], [4: (0.7608, 0.0374), (0.9728, 0.9177)], [5: (0.8921, 0.3680), (0.9779, 0.9486)], [6: (0.7813, 0.0565), (0.9702, 0.9002)], [7: (0.8942, 0.3786), (0.9711, 0.9062)], [8: (0.7848, 0.0605), (0.9611, 0.8280)].

A Levene test failed to reject the assumption of equal variances for every pair of sessions except the first (statistic, p-value): (6.3263, 0.0306), (2.2780, 0.1621), (1.2221, 0.2948), (0.8570, 0.3764), (2.7979, 0.1253), (0.7364, 0.4109), (0.0871, 0.7739), (0.0088, 0.9269).

Hence, we performed a two-sample independent *t*-test for every session with the following p-values: (0.4014, 0.04064, 0.0057, 0.0038, 0.0011, 0.0038, 0.0006, 6.3658e-05).

We also wished to test the difference between the slopes of linear fits to the accuracy curves for both conditions. A Shapiro-Wilk test failed to reject the assumption of normality (p<0.05) for either condition, with the following (W statistic, p-value) for 'above' and 'below': (0.8964, 0.3095), (0.8794, 0.3065). A Levene test failed to reject the assumption of equal variances (p<0.05) for each condition (statistic, p-value): (0.2141, 0.6535). Hence, we performed a two-sample independent *t*-test and found a significant difference (p=0.0027).

*Figure 6d*: We wished to test whether the animals had significantly changed their session mean $RTs$ with respect to their individual previous baseline $RTs$ (paired samples). To do this, we conducted a permutation test for every session with the new visible stimuli (stimulus pair 2) or the transparent stimuli. For 1000 repetitions, we randomly assigned labels to the experimental or baseline $RTs$ and then averaged the paired differences. The p-value for a particular session was the fraction of instances where the average permutation difference was more extreme than the actual experimental difference. For sessions with stimulus pair 2, the p-values from the permutation test were: (0.0034, 0.0069, 0.0165, 0.0071, 0.0291, 0.0347, 0.06, 0.0946, 0.3948, 0.244, 0.244, 0.4497, 0.3437). For sessions with transparent stimuli (plus rats AK2, AK9, and AK10 from the near-transparent stimuli), the p-values from the permutation were (0.0859375, 0.44921875, 0.15625, 0.03125, 0.02734375, 0.015625, 0.26953125, 0.02734375, 0.03125, 0.01953125, 0.0546875). To investigate whether the animals' significantly slowed down their mean $RTs$ compared to baseline during the first session of transparent stimuli, we divided $RTs$ in the first session in half and ran a permutation test on each half with the following p-values: (0.0390625, 0.2890625).

*Figure 6e*: In order to test the correlation between the initial change in $RT$ and the initial change in SNR for stimulus pair 2, we ran a standard linear regression on the average per subject for each of these variables for the first three sessions of stimulus pair 2 with $R^2 = 0.38$ and p-value = 0.01.

*Figure 2—figure supplement 1d—i*: Statistical significance of differences in means between the two training regimes for a variety of psychophysical measures was determined by a Wilcoxon rank-sum test with p<0.05. The p-values were: (**d**) accuracy: 0.21, (**e**) reaction time: 0.81, (**f**) fraction max *iRR*: 0.22, (**g**) total trial number: 0.46, (**h**): voluntary intertrial interval after error: 0.75, (**i**) fraction trials ignored: 0.03.

*Figure 4—figure supplement 2*: Statistical significance for mean parameters was calculated by taking the difference between the posterior distributions and using the proportion of the difference distribution that overlapped with 0 as the p-value. For individuals' parameters, p-values were determined via a Wilcoxon signed-rank test. The p-values were: (**a**) drift mean:<1e-4, drift individuals:<1e-4; (**b**) threshold mean: 0.0012, threshold individuals: 0.0008; (**c**) drift mean: <1e-4, drift individuals: <1e-4, threshold mean: 0.0298, threshold individuals: 0.0585; (**d**) drift mean: (baseline versus start learn: <1e-4, baseline versus after learn: 0.0378, start versus after learn: <1e-4), drift individuals: (baseline versus start learn: 0.0004, baseline versus after learn: 0.0703, start versus after learn: 0.0004); (**e**) threshold mean: (baseline versus start learn: 0.1100, baseline versus after learn: 0.3546, start versus after learn: 0.1904), threshold individuals: (baseline versus start learn: 0.0045, baseline versus after learn: 0.4380, start versus after learn: 0.1627); (**f**) drift mean: (baseline versus start learn: <1e-4, baseline versus after learn: 0.0616, start versus after learn: <1e-4), drift individuals: (baseline versus start learn: 0.0004, baseline versus after learn: 0.1089, start versus after learn: 0.0004), threshold mean: (baseline versus start learn: 0.2546, baseline versus after learn: 0.4614, start versus after learn: 0.2816), threshold individuals: (baseline versus start learn: 0.0200, baseline versus after learn: 0.7174, start versus after learn: 0.1089).

*Figure 4—figure supplement 3*: Statistical significance was determined as for *Figure 4—figure supplement 2*. The p-values were: (**a**) drift mean: <1e-4, drift individuals: <1e-4; (**b**) threshold mean: 0.0036, threshold individuals: 0.0013; (**c**) drift mean: <1e-4, drift individuals: <1e-4, threshold mean: 0.0314, threshold individuals: 0.0585; (**d**) drift mean: (baseline versus start learn: <1e-4, baseline versus after learn: 0.0428, start versus after learn: <1e-4), drift individuals: (baseline versus start learn: 0.0004, baseline versus after learn: 0.0703, start versus after learn: 0.0004); (**e**) threshold mean: (baseline versus start learn: 0.0866, baseline versus after learn: 0.4192, start versus after learn: 0.1252), threshold individuals: (baseline versus start learn: 0.0038, baseline versus after learn: 0.5349, start versus after learn: 0.1089); (**f**) drift mean: (baseline versus start learn: <1e-4, baseline versus after learn: 0.0618, start versus after learn: <1e-4), drift individuals: (baseline versus start learn: 0.0004, baseline versus after learn: 0.0980, start versus after learn: 0.0004), threshold mean: (baseline versus start learn: 0.2436, baseline versus after learn: 0.4596, start versus after learn: 0.2860), threshold individuals: (baseline versus start learn: 0.0200, baseline versus after learn: 0.7174, start versus after learn: 0.1089).

*Figure 4—figure supplement 4*: Statistical significance was determined as for *Figure 4—figure supplement 2*. The p-values were: (**a**) drift mean: <1e-4, drift individuals: <1e-4, drift variability: <1e-4; (**b**) threshold mean: 0.0036, threshold individuals:< 1e-4, drift variability: <1e-4; (**c**) drift mean: <1e-4, drift individuals: <1e-4, threshold mean: 0.0696, threshold individuals: 0.1587, drift variability: <1e-4; (**d**) drift mean: (baseline versus start learn: <1e-4, baseline versus after learn: 0.0422, start versus after learn: <1e-4), drift individuals: (baseline versus start learn: 0.0004, baseline versus after learn: 0.0557, start versus after learn: 0.0004), drift variability: (baseline versus start learn: 0.7940, baseline versus after learn: 0.8104, start versus after learn: 0.4564); (**e**) threshold mean: (baseline versus start learn: <1e-4, baseline versus after learn: 0.4188, start versus after learn: 0.0002), threshold individuals: (baseline versus start learn: 0.0004, baseline versus after learn: 0.5014, start versus after learn: 0.0004), drift variability: (baseline versus start learn: <1e-4, baseline versus after learn: 0.4442, start versus after learn: <1e-4); (**f**) drift mean: (baseline versus start learn:<1e-4, baseline versus after learn: 0.0596, start versus after learn: <1e-4), drift individuals: (baseline versus start learn: 0.0004, baseline versus after learn: 0.0980, start versus after learn: 0.0004), threshold mean: (baseline versus start learn: 0.2474, baseline versus after learn: 0.4702, start versus after learn: 0.2652), threshold individuals: (baseline versus start learn: 0.0200, baseline versus after learn: 0.7174, start versus after learn: 0.1089), drift variability: (baseline versus start learn: 0.2392, baseline versus after learn: 0.5294, start versus after learn: 0.2132).

*Figure 6—figure supplement 1a, b*: We tested for a difference in the aggregate reaction time distributions of a transparent stimuli condition in the first two and last two sessions ($n = 8$ subjects), and a no minimum reaction time condition with known stimuli ($n = 8$ subjects) via a two-sample Kolmogorov-Smirnov test and found a p-value of <1e-4 for both comparisons.

*Figure 6—figure supplement 3*: Statistical significance was determined as for *Figure 4—figure supplement 2*. The p-values were: (**a**) drift mean: (visible versus start transparent: 0.0002, visible versus end transparent: <1e-4, start versus end transparent: 0.3180), drift individuals: (visible versus

start transparent: 0.0117, visible versus end transparent: 0.0117, start versus end transparent: 0.5754), threshold mean: (visible versus start transparent: 0.1362, visible versus end transparent: 0.0094, start versus end transparent: 0.0658), threshold individuals: (visible versus start transparent: 0.0499, visible versus end transparent: 0.0173, start versus end transparent: 0.0173), drift variability: (visible versus start transparent: 0.1494, visible versus end transparent: 0.1614, start versus end transparent: 0.5032), $T_0$ mean: (visible versus start transparent: 0.0068, visible versus end transparent: 0.0194, start versus end transparent: 0.6106), $T_0$ individuals: (visible versus start transparent: 0.0117, visible versus end transparent: 0.0117, start versus end transparent: 0.0929).

*Figure 6—figure supplement 5b, d*: We tested for a difference in mean post-error slowing between the first two sessions and last two sessions of training for each animal for stimulus pair 1 (**b**) or the last two sessions of stimulus pair 1 and the first two sessions of stimulus pair 2 (**d**) via a Wilcoxon-signed rank test. The p-values were (**b**) 0.585 and (**d**) 0.255.

## Evaluation of optimality

Under the assumptions of a simple drift-diffusion process, the OPC defines a set of optimal threshold-to-drift ratios with corresponding decision times and error rates for which an agent maximizes instantaneous reward rate (*Bogacz et al., 2006*). Decision times are scaled by the particular task timing as mean normalized decision time: $DT/D_{err}$. The OPC is parameter free and can thus be used to compare performance across tasks, conditions, and individuals. An optimal agent will lie on different points on the OPC depending on differences in task timing ($D_{err}$) and stimulus difficulty (SNR). Assuming constant task timing, the SNR will determine different positions along the OPC for an optimal agent. For $DT > 0$ and $0 < ER < 0.5$, the OPC is defined as:

$$\frac{DT}{D_{err}} = \left[ \frac{1}{ER \log \frac{1-ER}{ER}} - \frac{1}{1 - 2ER} \right]^{-1} \tag{33}$$

and exists in speed-accuracy space, defined by $DT/D_{err}$ and ER. Given an estimate of $D_{err}$, the $ER$ and $DT$ for any given animal can be compared to the optimal values defined by the OPC in speed-accuracy space.

Moreover, because $ER$ should decrease with learning, learning trajectories for different subjects and models can also be compared to the OPC and to each other in speed-accuracy space.

Mean normalized decision time depends only on $D_{err}$.

For completeness, we include a derivation showing that the appropriate normalized decision time for the OPC depends only on $D_{err}$, not $D_{corr}$. According to *Gold and Shadlen, 2002*, average reward rate is defined as:

$$RR = \frac{\text{average accuracy}}{\text{average time per trial}}. \tag{34}$$

We can write the average reward rate as (see A26 from *Bogacz et al., 2006*),

$$RR = \frac{1 - ER}{DT + D_{corr} + ER(D_{err} - D_{corr})}. \tag{35}$$

Optimal behavior is defined as maximizing reward rate with respect to the thresholds in the DDM. We thus rewrite $ER$ and $DT$ in terms of average threshold and average SNR,

$$RR = \frac{1 - \frac{1}{1+e^{2\bar{z}\bar{A}}}}{\bar{z} \tanh\left(\bar{z}\bar{A}\right) + D_{corr} + \frac{1}{1+e^{2\bar{z}\bar{A}}}(D_{err} - D_{corr})} \tag{36}$$

$$= \frac{1}{\bar{z} + D_{corr} + (D_{err} - \bar{z})e^{-2\bar{z}\bar{A}}}. \tag{37}$$

Next to find the extremum, we take the derivative of $RR$ with respect to the threshold and set it to zero,

$$\frac{\partial RR}{\partial \bar{z}} = -\frac{1 + (-1 - 2\bar{A}(D_{err} - \bar{z}))e^{-2\bar{z}\bar{A}}}{(\bar{z} + D_{corr} + (D_{err} - \bar{z})e^{-2\bar{z}\bar{A}})^2} \tag{38}$$

$$0 = 1 - \left[1 + 2\bar{A}(D_{err} - \bar{z})\right] e^{-2\bar{z}\bar{A}} \tag{39}$$

$$= \frac{1 - 2ER}{ER} - \frac{D_{err}}{DT}(1 - 2ER)\log\frac{1 - ER}{ER} - \log\frac{1 - ER}{ER}, \tag{40}$$

where in the final step we have rewritten $\bar{z}$ and $\bar{A}$ in terms of $ER$ and $DT$.

Rearranging to place $DT$ on the left-hand side reveals an OPC where decision time is normalized by the post-error non-decision time $D_{err}$:

$$\frac{DT}{D_{err}} = \left[\frac{1}{ER\log\frac{1-ER}{ER}} - \frac{1}{1 - 2ER}\right]^{-1}. \tag{41}$$

Notably, the post-correct non-decision time $D_{corr}$ is not part of the normalization. Intuitively, this is because post-correct delays are an unavoidable part of accruing reward and therefore do not influence the optimal policy.

## Estimating $T_0$

$T_0$ is defined as the non-decision time component of a reaction time, comprising motor and perceptual processing time (**Holmes and Cohen, 2014**). It can be estimated by fitting a DDM to the psychophysical data. Because of the experimentally imposed minimum reaction time meant to ensure visual processing of the stimuli, however, our reaction time distributions were truncated at 350 ms, meaning a DDM fit estimate of $T_0$ is likely to be an overestimate. To address this issue, we set out to determine possible boundaries for $T_0$ and estimated it in a few ways, all of which did indeed fall between those boundaries (**Figure 1—figure supplement 4e**).

We found that after training, in the interval between 350 and 375 ms, nearly all of our animals had accuracy measurements above chance (**Figure 1—figure supplement 4b**), meaning that the minimum reaction time of 350 ms served as an upper bound to possible $T_0$ values.

To determine a lower bound, we obtained measurements for the two components comprising $T_0$: motor and initial perceptual processing times. To measure the minimum motor time required to complete a trial, we analyzed licking times across the different lickports. The latency from the last lick in the central port to the first lick in one of the two side ports peaked at around 80 ms (**Figure 1—figure supplement 4c**). In addition, the latency from one lick to the next lick at the same port at any of the lickports was also around 80 ms (data not shown). Because the latencies in lick times between lickports (requires movement of the head) and within the same lockport (does not require movement of the head) were about equal we concluded that the minimum motor time was determined by the limit on lick frequency, and not on a movement of the head redirecting the animal from the central port to one of the side ports. To measure the initial perceptual processing times, we looked to published latencies of visual stimuli traveling to higher visual areas in the rat. Published latencies reaching area TO (predicted to be after V1, LM, and LI in the putative ventral stream in the rat) were around 80 ms (**Figure 1—figure supplement 4d**; **Vermaercke et al., 2014**). Based on these measurements, we estimated a $T_0$ lower bound of approximately 160 ms.

One worry is that our lower bound could potentially be too low, as it is only estimated indirectly. Recent work on the SAT in a low-level visual discrimination tasks in rats found that accuracy was highest at a reaction time of 218 ms (**Kurylo et al., 2020**). However, accuracy was still above chance for reaction times binned between 130 and 180 ms. In this task, reaction time was measured when an infrared beam was broken, which means we can assume there was no motor processing time. This leaves decision time, and initial perceptual processing time (part of $T_0$) within the 130–180 ms duration. The complexity of solving a high-level visual task like ours and a low-level one will result in substantial differences in decision time, but should not in principle affect non-decision time. Considering a latency estimate of 80 ms based on physiological evidence (**Vermaercke et al., 2014**) can account for the initial perceptual processing component of $T_0$ and gives an estimate $T_0$=80 ms for this study.

Because a reaction time around $T_0$ should not allow for any decision time, accuracy should be around 50%. To estimate $T_0$ based on this observation, we extrapolated the time at which accuracy would drop to 50% after plotting accuracy as a function of reaction time (**Figure 1—figure supplement 4a**) and found values of 165 and 225 ms for linear and quadratic extrapolations respectively.

Finally, we fit our behavioral data with an HDDM (*Wiecki et al., 2013*) and found a $T_0$ estimate of 295 ±4 ms (despite there being no data below 350 ms). To address this issue, we fit a DDM to a small number of behavioral sessions we conducted with animals trained on the minimum reaction time of 350 ms but where that constraint was eliminated and found a $T_0$ estimate of 265 ± 120 (SD) ms. We stress that because the animals were trained with a minimum reaction time, they likely would have required extensive training without that constraint to fully make use of the time below the minimum reaction time, thus this estimate is likely to also be an overestimate. We do note however that the estimate is lower than the estimate with an enforced minimum reaction time and has a much higher standard deviation (spanning our lower and upper bound estimates).

Despite the range of possible $T_0$ values, we find that our qualitative findings (in terms of learning trajectory and near-optimality after learning) do not change (*Figure 1—figure supplement 4f, g*), and proceed with a $T_0$=160 ms for the main text.

## Determining $\tilde{D}_{err}$ and $\tilde{D}_{corr}$

The experimental protocol imposes a mandatory post-error and post-correct response-to-stimulus time ($\tilde{D}_{err}$ and $\tilde{D}_{corr}$, respectively). However, these times may not be accurate because of delays in the software communicating with different components such as the syringe pumps, and other delays such as screen refresh rates. We thus determined the actual mandatory post-error and post-correct response-to-stimulus times by measuring them based on timestamps on experimental file logs and found that $\tilde{D}_{err} = 3136$ ms, and $\tilde{D}_{corr} = 6370$ ms (*Figure 1—figure supplement 6*).

### Voluntary intertrial interval

We assume that the animals optimize reward rate based on task engagement time, that is, the sum of reaction times and all mandatory task delays, but not including any extra voluntary intertrial intervals. Therefore, our measures of non-decision task engagement time $D_{err}$ and $D_{corr}$ do not include voluntary intertrial intervals. In essence this amounts to the assumption that animals exit the task between trials, potentially pursuing other goals, and do not count this voluntary interval when measuring their within-task reward rate.

We conducted a detailed analysis of the voluntary intertrial intervals after both correct and error trials (*Figure 1—figure supplement 5*). To prevent a new trial from initiating while the animals were licking one of the side lickports, the task included a 300 ms interval at the end of a trial where an extra 500 ms were added if the animal licked one of the side lickports (*Figure 1—figure supplement 6*). There was no stimulus (visual or auditory) to indicate the presence of this task feature so the animals were not expected to learn it. It was clear that the animals did not learn this task feature as most voluntary intertrial intervals are clustered in 500 ms intervals and decay after each boundary (*Figure 1—figure supplement 5a*). Aligning the voluntary intertrial distributions every 500 ms reveals substantial overlap (*Figure 1—figure supplement 5c, d*), indicating similar urgency in every 500 ms interval, with an added amount of variance the farther the interval from zero. Moreover, measuring the median voluntary intertrial interval from 0 to 500, 0–1000, and 0–2000 ms showed very similar values (47, 67, 108 ms after error trials, *Figure 1—figure supplement 5b*). The median was higher after correct trials (55, 134, 512 ms, *Figure 1—figure supplement 5b*) because the animals were collecting reward from the side lickports and much more likely to trigger the extra 500 ms penalty times.

### Reward rate sensitivity to $T_0$ and voluntary intertrial interval

To ensure that our results did not depend on our chosen estimate for $T_0$ and our choice to ignore voluntary intertrial intervals when computing metrics like $D_{tot}$ and reward rate, we computed fraction maximum instantaneous reward rate as a function of $T_0$ and voluntary intertrial interval. We conducted this analysis across $n = 26$ rats at asymptotic performance (*Figure 1—figure supplement 7a, b*), and during the learning period (*Figure 1—figure supplement 7c, d*). During asymptotic performance, sweeping $T_0$ from our estimated minimum to our maximum possible values generated negligible changes in reward rate across a much larger range of possible voluntary intertrial intervals than we observed (*Figure 1—figure supplement 7a*). Reward rate was more sensitive to voluntary intertrial intervals, but did not drop below 90% of the possible maximum when considering a median voluntary intertrial interval up to 2000 ms (the median when allowing up to a 2000 ms window after a trial, after which agents are considered to have 'exited the task') (*Figure 1—figure supplement 7b*). During

learning, we found similar results, with possible voluntary intertrial interval values have a larger effect on reward rate than $T_0$, however even with the most extreme combination of a maximum $T_0$=350 ms, and the median voluntary intertrial interval up to 2000 ms (*Figure 1—figure supplement 7d*, light gray trace), fraction maximum reward rate was at most 10–15% away from the least extreme combination of $T_0$=160 ms and voluntary intertrial interval = 0 (*Figure 1—figure supplement 7c*, horizontal line along the bottom of the heat map) for most of the learning period. These results confirm that our qualitative findings do not depend on our estimated values of $T_0$ and choice to ignore voluntary intertrial intervals.

## Ignore trials

Because of the free-response nature of the task, animals were permitted to ignore trials after having initiated them (*Figure 1—figure supplement 2*). Although the fraction of ignored trials did seem to be higher at the beginning of learning for the first set of stimuli the animals learned (stimulus pair 1; *Figure 1—figure supplement 2a*), this effect did not repeat for the second set (stimulus pair 2, *Figure 1—figure supplement 2b*). This suggests that the cause for ignoring the trials during learning was not stimulus-based but rather related to learning the task for the first time. Overall, the mean fraction of ignored trials remained consistently low across stimulus sets and ignore trials were excluded from our analyses.

## Post-error slowing

In order to verify whether the increase in reaction time we saw at the beginning of learning relative to the end of learning was not solely attributable to a post-error slowing policy, we quantified the amount of post-error slowing during learning for both stimulus pair 1 and stimulus pair 2. For stimulus pair 1, we found that there was a consistent but slight amount of average post-error slowing (*Figure 6—figure supplement 5a*). This amount was not significantly different at the start and end of learning (*Figure 6—figure supplement 5b*).

We re-did this analysis for stimulus pair 2 and found similar results: animals had a consistent, modest amount of post-error slowing but it did not change across sessions during learning (*Figure 6—figure supplement 5c*). We tested for a significant difference in post-error slowing between the last two sessions of stimulus pair 1 and the first two sessions of the completely new stimulus pair 2 and found none (*Figure 6—figure supplement 5d*) even though there was a large immediate change in error rate. In fact, there was a trend toward a decrease in post-error slowing (and toward post-correct slowing) in the first few sessions of stimulus pair 2. This is consistent with the hypothesis that post-error slowing is an instance of a more general policy of orienting toward infrequent events (*Notebaert et al., 2009*). As correct trials became more infrequent than error trials when stimulus pair 2 was presented, we observed a trend toward post-correct slowing, as predicted by this interpretation.

Our subjects exhibit a modest, consistent amount of post-error slowing, which could at least partially explain the reaction time differences we see throughout learning. An experiment with transparent stimuli where error rate was constant but reaction times dropped, however, strongly contradicts the account that the rats implement a simple strategy like post-error slowing to modulate their reaction times during learning.

## RNN model and LDDM reduction

We consider a recurrent network receiving noisy visual inputs over time. In particular, we imagine that an input layer projects through weighted connections to a single recurrently connected read-out node, and that the weights must be tuned to extract relevant signals in the input. The read-out node activity is compared to a modifiable threshold which governs when a decision terminates. This network model can then be trained via error-corrective gradient descent learning or some other procedure. In the following we derive the average dynamics of learning.

To reduce this network to a DDM with time-dependent SNR, we first note that due to the law of large numbers, activity increments of the read-out node will be Gaussian provided that the distribution of input stimuli has bounded moments. We can thus model the input-to-readout pathway at each time step as a Gaussian input $x(t)$ flowing through a scalar weight $u$, with noise of variance $c_o^2$ added before the signal is sent into an integrating network. Taking the continuum limit, this yields a drift-diffusion process with effective drift rate $\tilde{A} = Au$ and noise variance $\tilde{c}^2 = u^2 c_i^2 + c_o^2$. Here, $A$ parameterizes the

perceptual signal, $c_i^2$ is the input noise variance (noise in input channels that cannot be rejected), and $c_o^2$ is the output noise variance (internal noise in output circuitry). The resulting decision variable $\hat{y}$ at time $T$ is Gaussian distributed as $N(AuTy, u^2c_i^2T + c_o^2T)$, where $y$ is the correct binary choice. A decision is made when $\hat{y}$ hits a threshold of $\pm z$.

## Within-trial drift-diffusion dynamics

On every trial, therefore, the subject's behavior is described by a drift-diffusion process, for which the average reward rate as a function of signal to noise and threshold parameters is known (***Bogacz et al., 2006***). The accuracy and decision time of this scheme is determined by two quantities. First, the SNR

$$\bar{A} = \left(\frac{\tilde{A}}{\tilde{c}}\right)^2 = \frac{A^2u^2}{u^2c_i^2 + c_o^2}, \tag{42}$$

and second, the threshold-to-drift ratio $\bar{z} = z/\tilde{A} = \frac{z}{Au(t)}$.

We can rewrite the SNR as

$$\bar{A}(t) = \frac{A^2u(t)^2}{c_i^2u(t)^2 + c_o^2} = \frac{A^2}{c_i^2 + c_o^2/u(t)^2}. \tag{43}$$

From this it is clear that, when learning has managed to amplify the input signals such that $u(t) \to \infty$, the asymptotic SNR is simply $\bar{A}^* = A^2/c_i^2$. Further, rearranging to

$$\bar{A}(t) = \frac{\bar{A}^*}{1 + \left(c_o^2/c_i^2\right)/u(t)^2} \tag{44}$$

shows that there are in fact just two parameters: the asymptotic achievable SNR $\bar{A}^*$ and the output-to-input noise variance ratio $c \equiv c_o^2/c_i^2$,

$$\bar{A}(t) = \frac{\bar{A}^*}{1 + c/u(t)^2}. \tag{45}$$

The mean error rate ($ER$), mean decision time ($DT$), and mean reward rate ($RR$) are therefore

$$ER = \frac{1}{1 + e^{2\bar{z}\bar{A}}} \tag{46}$$

$$DT = \bar{z}\tanh\left(\bar{z}\bar{A}\right) \tag{47}$$

$$RR = \frac{1 - ER}{DT + D_{tot}}, \tag{48}$$

where we have suppressed the dependence of $\bar{A}$ and $\bar{z}$ on time for clarity. Here, $D_{tot} = T_0 + ER \cdot D_{err} + (1 - ER) \cdot D_{corr}$ is the average non-decision task engagement time.

The term $\bar{z}\bar{A}$ is a measure of the total evidence accrued on average, and is equal to

$$\bar{z}\bar{A} = \frac{z}{Au(t)}\frac{\bar{A}^*}{1 + c/u(t)^2} \tag{49}$$

$$= \frac{z\bar{A}^*/A}{u(t) + c/u(t)}. \tag{50}$$

Here, for a fixed threshold $z$, the denominator shows the trade-off for increasing perceptual sensitivity: small $u(t)$ causes errors due to output noise, while large $u(t)$ causes errors due to overly fast integration for the specified threshold level.

## Across-trial error-corrective learning dynamics

To model learning, we consider that animals adjust perceptual sensitivities $u$ over time in service of minimizing an objective function. In this section we derive the average learning dynamics when the objective is to minimize the error rate. The LDDM can be conceptualized as an 'outer-loop' that modifies the SNR of a standard DDM 'inner-loop' described in the preceding subsection. If perceptual

learning is slow, there is a strong separation of timescales between these two loops. On the timescale of a single trial, the agent's SNR is approximately constant and evidence accumulation follows a standard DDM, whereas on the timescale of many trials, the specific outcome on any one trial has only a small effect on the network weights $w$, such that the learning-induced changes are driven by the *mean ER* and *DT*.

To derive the mean effect of error-corrective learning updates, we suppose that on each trial the network uses gradient descent on the hinge loss to update its parameters, corresponding to standard practice for supervised neural networks. The hinge loss is

$$\mathcal{L}(u, y) = \max(0, 1 - \hat{y}y), \tag{51}$$

yielding the gradient descent update.

$$u[r + 1] \leftarrow u[r] - \lambda \frac{\partial \mathcal{L}(u[r], y)}{\partial u}, \tag{52}$$

where $\lambda$ is the learning rate and $r$ is the trial number.

When the learning rate is small ($\lambda \ll 1$), each trial changes the weights minimally and the overall update is approximately given by the average continuous time dynamics

$$\frac{du}{dr} = -\left\langle \lambda \frac{\partial \mathcal{L}(u, y)}{\partial u} \right\rangle \tag{53}$$

$$= -\lambda \left\langle \left\langle \frac{\partial \mathcal{L}(u, y)}{\partial u} \middle| \text{error} \right\rangle + \left\langle \frac{\partial \mathcal{L}(u, y)}{\partial u} \middle| \text{correct} \right\rangle \right\rangle \tag{54}$$

$$= -\lambda ER \left\langle \frac{\partial \mathcal{L}(u, y)}{\partial u} \middle| \text{error} \right\rangle \tag{55}$$

$$= \lambda ER \left\langle y \frac{\partial \hat{y}}{\partial u} \middle| \text{error} \right\rangle, \tag{56}$$

where $\langle \cdot \rangle$ denotes an average over the correct answer $y$, the inputs and the output noise. The first step follows from iterated expectation. The second step follows from the fact that the probability of an error is simply the error rate $ER$, and for correct trials, the derivative of the hinge loss is zero. Next,

$$\frac{\partial \hat{y}}{\partial u} = \frac{\partial}{\partial u} \left( \sum_{i=0}^{T} u x_i + \eta_i \right) \tag{57}$$

$$= \sum_{i=0}^{T} x_i, \tag{58}$$

where $T$ is the time step at which $\hat{y}$ crosses the decision threshold $\pm z$. Returning to *Equation (56)*,

$$\lambda ER \left\langle y \frac{\partial \hat{y}}{\partial u} \middle| \text{error} \right\rangle = \lambda ER \left\langle y \sum_{i=1}^{T} x_i \middle| \text{error} \right\rangle. \tag{59}$$

Hence, the magnitude of the update depends on the typical total sensory evidence given that an error is made. To calculate this, let $\bar{x}_t = \sum_{i=0}^{t} x_i$ be the total sensory evidence up to time $t$, and $\bar{\eta}_t = \sum_{i=0}^{t}$ be the total decision noise up to $t$. These are independent and normally distributed as

$$\bar{x}_t \sim N\left( y A t dt, c_i^2 t dt \right) \tag{60}$$

$$\bar{\eta}_t \sim N\left( 0, c_o^2 t dt \right). \tag{61}$$

Therefore, we have

$$\left\langle y \sum_{i=1}^{T} x_i \middle| \text{error} \right\rangle = \left\langle y \bar{x}_T \middle| \text{error} \right\rangle \tag{62}$$

$$= \left\langle y\bar{x}_T \middle| u\bar{x}_T + \bar{\eta}_T = -yz \right\rangle \tag{63}$$

$$= \left\langle y\bar{x}_T \middle| u\bar{x}_T/y + \bar{\eta}_T/y = -z \right\rangle. \tag{64}$$

These variables are jointly Gaussian. Letting $v_1 = y\bar{x}_T$ and $v_2 = u\bar{x}_T/y + \bar{\eta}_T/y$, the means $\mu_1, \mu_2$, variances $\sigma_1^2, \sigma_2^2$, and covariance $\mathrm{Cov}(v_1, v_2)$ of $v_1, v_2$ given the hitting time $T$ are

$$\mu_1 = ATdt \tag{65}$$

$$\mu_2 = uATdt \tag{66}$$

$$\sigma_1^2 = c_i^2 Tdt \tag{67}$$

$$\sigma_2^2 = u^2 c_i^2 Tdt + c_o^2 Tdt \tag{68}$$

$$\mathrm{Cov}(y\bar{x}_T, u\bar{x}_T/y + \bar{\eta}_T/y) = \langle y\bar{x}_T(u\bar{x}_T/y + \bar{\eta}_T/y)\rangle - \langle y\bar{x}_T\rangle\langle u\bar{x}_T/y + \bar{\eta}_T/y\rangle \tag{69}$$

$$= u\langle \bar{x}_T^2\rangle - u\langle \bar{x}_T\rangle^2 \tag{70}$$

$$= u c_i^2 Tdt. \tag{71}$$

The conditional expectation is therefore

$$\langle\langle v_1 | v_2 = -z, T\rangle\rangle_{T|\mathrm{error}} = \left\langle \mu_1 + \frac{\mathrm{Cov}(v_1, v_2)}{\sigma_2^2}(-z - \mu_2)\right\rangle_{T|\mathrm{error}} \tag{72}$$

$$= \left\langle ATdt + \frac{uc_i^2}{u^2 c_i^2 + c_o^2}(-z - uATdt)\right\rangle_{T|\mathrm{error}} \tag{73}$$

$$= A(DT) - \frac{uc_i^2}{u^2 c_i^2 + c_o^2}(z + uA(DT)), \tag{74}$$

where we have used the fact that $\langle Tdt\rangle_{T|\mathrm{error}} = DT$, because in the DDM the mean decision time is the same for correct and error trials. Inserting *Equation (74)* into *Equation (59)* yields

$$\frac{d}{dr}u = \lambda ER\left(A(DT) + \frac{1}{1 + \frac{c_o^2}{u^2 c i^2}}(-z/u - A(DT))\right) \tag{75}$$

$$= \lambda ER\left(A(DT) - \frac{1}{1 + c/u^2}(z/u + A(DT))\right). \tag{76}$$

Finally, we switch the units of the time variable from trials to seconds using the relation $dt = (DT + D_{tot})dr$, yielding the dynamics

$$\tau\frac{d}{dt}u = \frac{ER}{DT + D_{tot}}\left(A(DT) - \frac{1}{1 + c/u^2}\left[\frac{z}{u} + A(DT)\right]\right). \tag{77}$$

The above equation describes the dynamics of $u$ under gradient descent learning. We note that here, the dependence of the dynamics on threshold trajectory is contained implicitly in the $DT$, $ER$, and $D_{tot}$ terms.

To obtain equivalent dynamics for the SNR $\bar{A}$, we have

$$\tau\frac{d}{dt}\bar{A} = 2\frac{A^2 c_o^2 u}{(c_i^2 + c_o^2/u(t)^2)^2}u(t)^{-3}\frac{d}{dt}u \tag{78}$$

$$= 2\frac{c}{\bar{A}^*}\bar{A}^2 u^{-3}\frac{d}{dt}u. \tag{79}$$

Rearranging the definition of $\bar{A}$ yields

$$u^2 = \frac{c\bar{A}}{\bar{A}^* - \bar{A}}. \tag{80}$$

Inserting (*Equation 80*) into (*Equation 79*) and simplifying, we have

$$\tau \frac{d}{dt}\bar{A} = 2\sqrt{\frac{\bar{A}\left(\bar{A}^*\right)}{c}} \left(1 - \frac{\bar{A}}{\bar{A}^*}\right)^{3/2} \frac{d}{dt}u \tag{81}$$

$$= 2\sqrt{\frac{\bar{A}\left(\bar{A}^*\right)}{c}} \left(1 - \frac{\bar{A}}{\bar{A}^*}\right)^{3/2} \frac{ER}{DT + D_{tot}} \left(A(DT) - \frac{1}{1 + c/u^2}\left[\frac{z}{u} + A(DT)\right]\right) \tag{82}$$

$$= 2A\sqrt{\frac{\bar{A}\left(\bar{A}^*\right)}{c}} \left(1 - \frac{\bar{A}}{\bar{A}^*}\right)^{5/2} \frac{ER}{DT + D_{tot}} \left[DT - \frac{\log(1/ER - 1)}{\bar{A}^*\left(1 - \frac{\bar{A}}{\bar{A}^*}\right)^2}\right]. \tag{83}$$

Here, in the second step we have used the fact that $\bar{A} = \frac{1-2ER}{2\langle DT \rangle} \log \frac{1-ER}{ER}$ and *Equation (80)*. Finally, absorbing the drift rate $A$ into the time constant $\tau = \frac{1}{A\lambda}$, we have the dynamics

$$\tilde{\tau}\frac{d}{dt}\bar{A} = 2\sqrt{\frac{\bar{A}(\bar{A}^*)}{c}} \left(1 - \frac{\bar{A}}{\bar{A}^*}\right)^{5/2} \frac{ER}{DT+D_{tot}} \left[DT - \frac{\log(1/ER-1)}{\bar{A}^*\left(1 - \frac{\bar{A}}{\bar{A}^*}\right)^2}\right]. \tag{84}$$

This equation reveals that the LDDM has four scalar parameters: the asymptotic SNR $\bar{A}^*$, the output-to-input-noise variance ratio $c$, the initial SNR at time zero $\bar{A}(0)$, and the combined drift rate/learning rate time constant $\tilde{\tau}$. In addition, it requires the choice of threshold trajectory $z(t)$.

To reveal the basic learning speed/instantaneous reward rate trade-off in this model, we investigate the limit where $\bar{A}$ is small but finite (low signal-to-noise) and the threshold is small, such that the error rate is near $ER = 1/2$. Then the final term in *Equation (84)* goes to zero, giving

$$\tilde{\tau}\frac{d}{dt}\bar{A} \approx \sqrt{\frac{\bar{A}\left(\bar{A}^*\right)}{c}} \left(1 - \frac{\bar{A}}{\bar{A}^*}\right)^{5/2} \frac{DT}{DT + D_{tot}} \tag{85}$$

$$\propto \frac{DT}{DT + D_{tot}}, \tag{86}$$

such that learning speed is increasing in $DT$. By contrast the instantaneous reward rate when $ER = 1/2$ is

$$RR \approx \frac{1/2}{DT + D_{tot}}, \tag{87}$$

which is a decreasing function of $DT$.

We note that when the perceptual signal is small, $DT$ is determined by the ratio of threshold to diffusion noise. Starting with *Equation 47*, we rewrite it in terms of threshold, perceptual signal, and noise:

$$DT = \frac{z}{\tilde{A}} \tanh\left(\frac{z}{\tilde{A}}\frac{\tilde{A}^2}{\tilde{c}^2}\right). \tag{88}$$

If we explore the limit in which perceptual signal is small, and following L'Hôpital's rule:

$$\lim_{\tilde{A}\to 0} DT = \lim_{\tilde{A}\to 0} \frac{\frac{d}{d\tilde{A}}z\tanh(\frac{z}{\tilde{c}^2}\tilde{A})}{\frac{d}{d\tilde{A}}\tilde{A}} = \lim_{\tilde{A}\to 0} \frac{z\,\text{sech}^2(\frac{z}{\tilde{c}^2}\tilde{A})}{1} \tag{89}$$

Leaving:

$$\lim_{\tilde{A}\to 0} DT = \left(\frac{z}{\tilde{c}}\right)^2. \tag{90}$$

Thus, a change in $DT$ when perceptual signal is low could be caused by either a changing threshold with fixed diffusion noise, a constant threshold with varying diffusion noise, or a combination thereof, without the immediate ability to tell these apart. In these cases, however, we note that the ratio of threshold to diffusion noise cannot stay constant if $DT$ changes.

## Threshold policies

We evaluate several simple threshold policies.

The *iRR*-greedy policy sets $\bar{z} = \bar{z}^*$, the instantaneous reward rate optimal policy at all times.

The constant threshold policy sets $\bar{z}$ to a fixed constant throughout learning.

The *iRR*-sensitive policy implements a threshold $z^s(t)$ that decays with time constant $\gamma$ from an initial value $z^s(0) = z_0$ toward the *iRR*-optimal threshold,

$$\gamma \frac{d}{dt} z^s(t) = z^*(\bar{A}(t)) - z^s(t). \tag{91}$$

where $\gamma$ controls the rate of convergence.

Finally, the global optimal policy optimizes the entire function $\bar{z}(t)$ to maximize total cumulative reward during exposure to the task. To compute the optimal threshold trajectory, we discretize the reduction dynamics in *Equation 77* and perform gradient ascent on $\bar{z}(t)$ using automatic differentiation in the PyTorch python package. While this procedure is not guaranteed to find the global optimum (due to potential nonconvexity of the optimization problem), in practice we found highly reliable results from a range of initial conditions and believe that the identified threshold trajectory is near the global optimum.

## Parameter fitting

The LDDM has several parameters governing its performance, including the asymptotic optimal SNR, the output/input noise variance ratio, the learning rate, and parameters controlling threshold policies where applicable. To fit these, we discretized the reduction dynamics and performed gradient ascent on the log likelihood of the observed data under the LDDM, again using automatic differentiation in the PyTorch python package. Because our model is highly simplified, our goal was only to place the parameters in a reasonable regime rather than obtain quantitative fits. We note that our fitting procedure could become stuck in local minima, and that a range of other parameter settings might also be consistent with the data. The best fitting parameters we obtained and used in all model results were $A = 0.9542, c_i = 0.3216, c_o = 30, u_0 = 0.0001$. We used a discretization time step of $dt = 160$. For the constant threshold and *iRR*-sensitive policies, the best fitting initial threshold was $z(0) = 30$. For the *iRR*-sensitive policy, the best fitting decay rate was $\gamma = 0.00011891$.

## Acknowledgements

We thank Chris Baldassano, Christopher Summerfield, Rahul Bhui, Grigori Guitchounts, Laura Bustamante, Sebastian Musslick, and Jonathan D Cohen for useful discussions. We thank Joshua Breedon for summer assistance in developing faster animal training procedures. We thank Ed Soucy and the NeuroTechnology Core for help with improvements to the behavioral response rigs. This work was supported by the Richard A and Susan F Smith Family Foundation and IARPA contract # D16PC00002. JM was supported by the Harvard Brain Science Initiative (HBI) and the Department of Molecular and Cellular Biology at Harvard, and a Presidential Postdoctoral Research Fellowship at Princeton. AMS was supported by a Swartz Postdoctoral Fellowship in Theoretical Neuroscience and a Sir Henry Dale Fellowship from the Wellcome Trust and Royal Society (Grant Number 216386/Z/19/Z).

## Additional information

### Funding

| Funder | Grant reference number | Author |
| --- | --- | --- |
| Intelligence Advanced Research Projects Activity | D16PC00002 | David D Cox |
| Richard and Susan Smith Family Foundation | | David D Cox |

| Funder | Grant reference number | Author |
|--------|------------------------|--------|
| Harvard University | Harvard Brain Science Initiative | Javier Masís |
| Princeton University | Presidential Postdoctoral Research Fellowship | Javier Masís |
| Royal Society | Sir Henry Dale Fellowship (216386/Z/19/Z) | Andrew M Saxe |
| Wellcome Trust | Sir Henry Dale Fellowship (216386/Z/19/Z) | Andrew M Saxe |
| Swartz Foundation | Swartz Postdoctoral Fellowship in Theoretical Neuroscience | Andrew M Saxe |

The funders had no role in study design, data collection and interpretation, or the decision to submit the work for publication. For the purpose of Open Access, the authors have applied a CC BY public copyright license to any Author Accepted Manuscript version arising from this submission.

### Author contributions

Javier Masís, Conceptualization, Formal analysis, Investigation, Methodology, Project administration, Software, Validation, Writing – original draft, Writing – review and editing, Conceived the work; Travis Chapman, Software, Methodology, Aided JM in establishing initial operant training procedures and behavioral analysis; Juliana Y Rhee, Resources, Methodology, Designed the behavioral response rigs; David D Cox, Resources, Supervision, Funding acquisition, Methodology, Provided input to experimental design and acquired funding for the project; Andrew M Saxe, Conceptualization, Formal analysis, Methodology, Project administration, Software, Supervision, Visualization, Writing – original draft, Writing – review and editing, Conceived the work

### Author ORCIDs

Javier Masís (ID) http://orcid.org/0000-0002-9643-8677
Andrew M Saxe (ID) http://orcid.org/0000-0002-9831-8812

### Ethics

All care and experimental manipulation of animals were reviewed and approved by the Harvard Institutional Animal Care and Use Committee (IACUC), protocol 27-22.

### Decision letter and Author response

Decision letter https://doi.org/10.7554/eLife.64978.sa1
Author response https://doi.org/10.7554/eLife.64978.sa2

## Additional files

### Supplementary files

• Transparent reporting form

### Data availability

The data and code are freely available at https://github.com/jmasis/strategiclearning_and_lddm (copy archived at swh:1:rev:26aa21d1e830657896325b1a26e9b84f5e3be93d).

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
