## [Editor Report]

This manuscript provides a fresh view on the fundamental trade-off between the speed and accuracy of perceptual decision-making. Using computational modeling, the authors establish the important finding that adopting a momentary suboptimal trade-off for maximizing reward rate at the beginning of learning can yield better decisions and larger rewards at later stages. This novel prediction is tested in rodent experiments. The experiments and their detailed analysis provide compelling evidence for the authors' theoretical predictions.

---

## [Decision Letter]

**Decision letter after peer review:**

Thank you for submitting your article "Strategically managing learning during perceptual decision making" for consideration by *eLife*. Your article has been reviewed by 2 peer reviewers, and the evaluation has been overseen by Valentin Wyart as the Reviewing Editor and Michael Frank as the Senior Editor. The following individual involved in review of your submission has agreed to reveal their identity: Konstantinos Tsetsos (Reviewer #1).

The reviewers have discussed the reviews with one another and the Reviewing Editor has drafted this decision to help you prepare a revised submission.

Classical descriptions of the speed-accuracy trade-off during perceptual decision-making assume that agents balance decision speed and accuracy given a fixed level of perceptual sensitivity. These descriptions ignore how agents learn to process the incoming sensory information for the purpose of decision-making. This manuscript develops a theory for how this perceptual learning ought to occur, and tests predictions from the theory using rodent experiments. This theory of perceptual learning leads to a new way of understanding suboptimal slow decisions at the early stages of learning. The manuscript is theoretically and technically sound. Additionally, the experiments are ingeniously designed and rigorously analysed, and their results provide empirical support for the theory proposed by the authors. There are however additional analyses that should be performed to validate the authors' specific claims regarding the strategic adaptation of perceptual sensitivity throughout task execution. Furthermore, the manuscript could be improved for clarity.

A current weakness of the manuscript concerns the preference for a strategic adaptation of perceptual sensitivity throughout learning (iRR-sensitive policy) over a simpler gradual increase in perceptual sensitivity (constant-threshold policy). Indeed, the strategies provide qualitatively similar predictions (see, e.g., Figure 3). Validating the iRR-sensitive policy over the constant-threshold policy is critical to the overall conclusions of the work (namely, that rats are strategically adapting their decision times to promote learning). However, it is currently unclear how the constant-threshold policy (in which rats do not control the speed of their decision but benefit from a gradual improvement in perceptual sensitivity) has been conclusively ruled out.

To rule out the constant-threshold policy, the authors argue first that drift-diffusion model (DDM) fits to the data show both drift rate and decision boundary changes throughout learning (Figure S5). However, it is not clear that a concurrent change in drift rate and decision boundary is the most parsimonious explanation of the data. The authors should establish that indeed both parameters change via comparing different versions of the DDM where a subset of the parameters are allowed to vary while others remain fixed throughout learning. Additionally, it would be interesting to know whether the conclusions remain the same when a more 'complete' version of the DDM is used (including drift-rate variability as a free parameter).

The second argument in support of the iRR-sensitive policy comes from experiment 2, in which the authors convincingly show that the improvement in perceptual sensitivity (or SNR) scales with decision times. It is indeed important to show that longer viewing leads to larger SNR improvements. However, it is currently unclear how this observation rules out the constant-threshold policy. Unless additional analyses are performed to show that the constant-threshold policy does not make this prediction, this observation appears necessary but not sufficient to validate the iRR-sensitive policy.

The third argument in support of the iRR-sensitive policy comes from experiment 3, in which a first group of rats performed a 'learnable' perceptual experiment while a second group performed an experiment with 'unlearnable' (transparent) stimuli. Indeed, this second group showed a reduction in reaction times as the experiment progressed, which (a) is reward-rate optimal, and (b) can be understood as a strategic change of decision boundary, since the SNR in this experiment is theoretically zero. However, it is unclear how rodents behave in this 'unlearnable' context. Presumably, during the first two sessions, the rats may be trying to figure out what the task is (e.g., waiting to see if there are visible stimuli in a subset of trials). In the third session, the rats speed up but it is not clear if they keep speeding up later in the experiment. Are reaction times significantly decreasing beyond the third session? Finally, it is not obvious that the effective SNR in this experiment is zero. In this 'unlearnable' experiment, rats may use some non-sensory information (e.g., choice history information such as their preceding response and whether they got rewarded) as input to their drift rate.

Another weakness comes from the use of recurrent neural networks (RNNs) to model that accumulation of decision-relevant evidence. Indeed, these networks are tuned such that they become equivalent to DDMs. Framed in this way, the connection between RNNs and DDMs appears somewhat trivial, such that the introduction of RNNs does not add anything to the manuscript, and might even be confusing to some readers. The authors should either reframe the specific role of RNNs for supporting their key findings, or possibly remove them if they do not provide unique insights beyond classical drift-diffusion modeling.

The detailed description of the models is currently hidden in the Methods section, even though it is essential for understanding their learning dynamics. In particular, the authors assume two sources of noise in the model: one on the input, and one on the accumulator. Learning is achieved by re-scaling the input by an 'input weight'. Increasing the input weight boosts the input signal compared to the accumulation noise, such that the latter can be effectively suppressed by making the input weight large enough. By contrast, the input noise cannot be suppressed by such re-scaling, such that it is this noise that ultimately limits asymptotic performance, and determines the asymptotic SNR. This important constraint is currently not clear after reading the manuscript. The authors should reframe the manuscript to highlight and discuss the model variables that affect the SNR, including the input weight. This would clarify what the authors mean by 'learning' in their theory. The authors initialize the model with a low input weight to reflect that the agent has not yet learned how to interpret the sensory information for the purpose of decision-making. Thus, the input weight is not something that would have a direct mechanistic implementation in the brain. Instead, it is an abstract quantity that describes how well a decision-maker can turn sensory information into a perceptual decision. Once this interpretation of input weight is stated clearly in the manuscript (which it is not in the current version), starting the task with a low input weight makes sense.

Finally, a few choices made by the authors are critical for their findings, but are not sufficiently described in the main text. First, the observed reaction time (RT) is composed of a non-decision time (T0) and a decision time (DT). Experiments allow to measure RT, but the theory makes predictions about DT, which requires inferring T0. The magnitude of T0 impacts the results, but how it is inferred is currently buried in the Methods section. Second, the rodents sometimes choose to not immediately initiate a new trial (the "voluntary inter-trial interval"). The authors assume that rodents ignore this interval when maximizing reward rate, and find near-optimal reward rates under this assumption. Importantly, including this voluntary inter-trial interval makes reward rates drop significantly. However, these details are again buried in the Methods section, and are not mentioned nor discussed in the main text.

– The main conclusions hinge upon concurrent changes in both drift rate and decision boundary throughout learning. This change is assessed via fitting HDDM models. It is important that the authors fit and compare the following 3 DDM variants: (a) drift rate changes throughout learning but decision boundary remains fixed, (b) decision boundary changes throughout learning but drift rate remains fixed, (c) both drift rate and decision boundary change throughout learning.

– The HDDM fitting procedure is not fully described. In particular, it is not clear what parameters, asides the drift rate, the decision boundary and the non-decision time, varied. For instance, was the drift rate variability a free parameter? It is important to report more precisely the details of the model fits and, if simple variants of the DDM were used, to fit the data using more complex DDMs that include drift rate variability as a free parameter. Additionally, please report in Figure S5 all parameter estimates during learning and not just the drift rate and the decision boundary. Details around Equation (40) should also be in the main text. Furthermore, except for the internal noise, the setup looks very similar to that of Drugowitsch et al. (2019), and the relationship should be discussed somewhere in the main text.

– Currently the evolution of the mean RT during learning is examined. Plotting the change in the RT distribution (averaged/vincentised across participants) can be more informative about changes in the strategy being used than plotting the mean RT alone.

– The results of experiment 3 are interesting, but at the same time the behaviour in the transparent group requires more scrutiny. How do rats behave in this condition? It appears that random choice in combination with heuristic strategies (e.g. win-stay lose-switch) are viable possibilities. The argument that the SNR is intrinsically zero in this task. However, the signal in this experiment may contain non-zero, irrelevant information (such as choice history or feedback information). Plotting the RT distributions as a function of learning could provide insight in this regard, because boundary changes and SNR changes manifest differently in the shapes of RT distributions.

– The use of RNNs is not sufficiently supported beyond classical drift-diffusion modeling. The authors should either reframe the specific role of RNNs for supporting their key findings, or possibly remove them if they do not provide unique insights beyond classical drift-diffusion modeling.

– The authors should clarify early on in the manuscript what "learning" means in their model and theory, and why it makes sense to start with a low input weight (after describing the meaning of the input weight). The authors should also explain what T0 and D_RSI are, how they are determined (and the choices made to determine them), and how these parameters impact the results (also related to Figure S13). In particular the relationship between RT and DT is already required to understand Figure 1c, and many plots thereafter.

– Box 1 provides some details of the model, but leaves out others – e.g., the different sources of noise in the model. From Box 1 alone, it is unclear how the asymptotic SNR or the iRR-sensitive threshold are computed. It is indeed nice that it is possible to derive Equation (3), but the equation itself is not particularly informative for the exposition of the main findings, and so it could be moved to the Methods section.

---

## [Author Response]

The reviewers have discussed the reviews with one another and the Reviewing Editor has drafted this decision to help you prepare a revised submission.1) Classical descriptions of the speed-accuracy trade-off during perceptual decision-making assume that agents balance decision speed and accuracy given a fixed level of perceptual sensitivity. These descriptions ignore how agents learn to process the incoming sensory information for the purpose of decision-making. This manuscript develops a theory for how this perceptual learning ought to occur, and tests predictions from the theory using rodent experiments. This theory of perceptual learning leads to a new way of understanding suboptimal slow decisions at the early stages of learning. The manuscript is theoretically and technically sound. Additionally, the experiments are ingeniously designed and rigorously analysed, and their results provide empirical support for the theory proposed by the authors. There are however additional analyses that should be performed to validate the authors' specific claims regarding the strategic adaptation of perceptual sensitivity throughout task execution. Furthermore, the manuscript could be improved for clarity.A current weakness of the manuscript concerns the preference for a strategic adaptation of perceptual sensitivity throughout learning (iRR-sensitive policy) over a simpler gradual increase in perceptual sensitivity (constant-threshold policy). Indeed, the strategies provide qualitatively similar predictions (see, e.g., Figure 3). Validating the iRR-sensitive policy over the constant-threshold policy is critical to the overall conclusions of the work (namely, that rats are strategically adapting their decision times to promote learning). However, it is currently unclear how the constant-threshold policy (in which rats do not control the speed of their decision but benefit from a gradual improvement in perceptual sensitivity) has been conclusively ruled out.To rule out the constant-threshold policy, the authors argue first that drift-diffusion model (DDM) fits to the data show both drift rate and decision boundary changes throughout learning (Figure S5). However, it is not clear that a concurrent change in drift rate and decision boundary is the most parsimonious explanation of the data. The authors should establish that indeed both parameters change via comparing different versions of the DDM where a subset of the parameters are allowed to vary while others remain fixed throughout learning. Additionally, it would be interesting to know whether the conclusions remain the same when a more 'complete' version of the DDM is used (including drift-rate variability as a free parameter).

We have fit the data with several different versions of the DDM where a subset of the parameters are allowed to vary with learning while others remain fixed, including versions with drift rate variability, as requested by the reviewers (Figure 4—figure supplements 9-12). We found that the best model fits were those where both drift rate and threshold were allowed to vary with learning, and the presence of drift rate variability did not seem to noticeably improve model fits (Figure 4—figure supplement 9).

The second argument in support of the iRR-sensitive policy comes from experiment 2, in which the authors convincingly show that the improvement in perceptual sensitivity (or SNR) scales with decision times. It is indeed important to show that longer viewing leads to larger SNR improvements. However, it is currently unclear how this observation rules out the constant-threshold policy. Unless additional analyses are performed to show that the constant-threshold policy does not make this prediction, this observation appears necessary but not sufficient to validate the iRR-sensitive policy.

The reviewers are correct that this experiment does not rule out the constant-threshold policy, and therefore in the text we did not make this claim. The purpose of this experiment was solely to verify that longer viewing times lead to larger SNR improvements (a key prediction of our learning framework regardless of threshold policy).

The third argument in support of the iRR-sensitive policy comes from experiment 3, in which a first group of rats performed a 'learnable' perceptual experiment while a second group performed an experiment with 'unlearnable' (transparent) stimuli. Indeed, this second group showed a reduction in reaction times as the experiment progressed, which (a) is reward-rate optimal, and (b) can be understood as a strategic change of decision boundary, since the SNR in this experiment is theoretically zero. However, it is unclear how rodents behave in this 'unlearnable' context. Presumably, during the first two sessions, the rats may be trying to figure out what the task is (e.g., waiting to see if there are visible stimuli in a subset of trials). In the third session, the rats speed up but it is not clear if they keep speeding up later in the experiment. Are reaction times significantly decreasing beyond the third session? Finally, it is not obvious that the effective SNR in this experiment is zero. In this 'unlearnable' experiment, rats may use some non-sensory information (e.g., choice history information such as their preceding response and whether they got rewarded) as input to their drift rate.

The reviewers make an astute observation that the effective SNR for the group experiencing the transparent stimuli may not be zero due to stimulus-independent information, such as choice and reward history. To test for these strategies, we fit a generalized linear model to trial-bytrial choices for every subject and found that overall there was an increase in the weights associated with these strategies (perseverance and win-stay/lose-switch) (Figure 6—figure supplement 4). HDDM fits for this experiment also indicated that there appeared to be a drift rate greater than 0, indicating a non-zero SNR, perhaps due to these stimulus-independent strategies (Figure 6—figure supplement 3). However, the HDDM fits still found a decrease in threshold, as predicted by our model, and did not find an increase in drift rate (Figure 6—figure supplement 3). These threshold and drift rate trajectories differ from the ones found in the learnable stimuli, indicating that although the rats could be implementing stimulus-independent “monitoring” strategies, their choice of threshold still indicated a reduced belief in stimulus learnability. Overall, we find that the presence of these strategies does not invalidate our conclusions. We also make a terminological distinction: the SNR may refer to the signal in the task that is predictive of the correct answer. On this definition, because the correct answer is random in this experiment, the SNR is zero regardless of any information employed by the animal, as no strategy will attain better than 50% accuracy. Said another way, in the DDM, if drift rate is drift toward the correct answer, then this can only be zero. However, in the way used by the reviewer, SNR could correspond to a more mechanistic account based on the DDM and reflect signals supplied to the integrator even if uncorrelated with the rewarded side on a trial. These new analyses focus on this second definition, but we believe our point about speeding up responses because of a lack of task-relevant information can still be appreciated using the first definition.

Another weakness comes from the use of recurrent neural networks (RNNs) to model that accumulation of decision-relevant evidence. Indeed, these networks are tuned such that they become equivalent to DDMs. Framed in this way, the connection between RNNs and DDMs appears somewhat trivial, such that the introduction of RNNs does not add anything to the manuscript, and might even be confusing to some readers. The authors should either reframe the specific role of RNNs for supporting their key findings, or possibly remove them if they do not provide unique insights beyond classical drift-diffusion modeling.

We thank the reviewer for pointing to this important lack of clarity in the text. The RNN is essential to the analysis, because it is only from the RNN that the learning dynamics for the DDM parameters can be derived. The derivation of the differential equation on the SNR (Equation 8) is a central result of our work, and relies on the RNN interpretation. Starting with the RNN, we derive the effect of doing standard gradient based learning of the underlying weight parameters. We then back out the SNR dynamics in a DDM that exactly correspond to these gradient based updates. With only the DDM, it is not clear what differential equation should govern changes in SNR due to learning. Therefore, the main result that learning speed scales with viewing time is derived from the RNN interpretation and cannot be seen just from a DDM framework. Said another way, while the RNN and DDM are trivially the same for the within-trial dynamics where parameters are constant, by itself the DDM makes no predictions for across-trial learning dynamics, where parameters change as a function of mean performance. For this, the RNN interpretation is required. We note that gradient descent is not invariant to parametrization and so performing gradient descent directly on the SNR variable of a DDM is not equivalent to the SNR dynamics we have derived based on gradient descent on the weights in the underlying RNN. We have expanded the model explanation in the main text considerably, and included the following sentences highlighting the necessity of an RNN for our reduction to a DDM:

“Remarkably, this reduction shows that the high-dimensional dynamics of the RNN receiving stochastic pixel input and performing gradient descent on the weights (Figure 3, grey trace) can be described by a drift diffusion model with a single deterministic scalar variable--the effective SNR--that changes over time (Figure 3, blue trace). Notably, without the mapping to the original recurrent neural network, it is not possible to understand what effect error-corrective gradient descent learning would have at the level of the DDM, or how the learning process is influenced by choice of decision times. In particular, the change in SNR that arises from gradient descent on the underlying RNN weights (Equation10) is not equivalent to that arising from gradient descent on the SNR parameter in the DDM directly because gradient descent is not parametrization invariant.”

The detailed description of the models is currently hidden in the Methods section, even though it is essential for understanding their learning dynamics. In particular, the authors assume two sources of noise in the model: one on the input, and one on the accumulator. Learning is achieved by re-scaling the input by an 'input weight'. Increasing the input weight boosts the input signal compared to the accumulation noise, such that the latter can be effectively suppressed by making the input weight large enough. By contrast, the input noise cannot be suppressed by such re-scaling, such that it is this noise that ultimately limits asymptotic performance, and determines the asymptotic SNR. This important constraint is currently not clear after reading the manuscript. The authors should reframe the manuscript to highlight and discuss the model variables that affect the SNR, including the input weight. This would clarify what the authors mean by 'learning' in their theory. The authors initialize the model with a low input weight to reflect that the agent has not yet learned how to interpret the sensory information for the purpose of decision-making. Thus, the input weight is not something that would have a direct mechanistic implementation in the brain. Instead, it is an abstract quantity that describes how well a decision-maker can turn sensory information into a perceptual decision. Once this interpretation of input weight is stated clearly in the manuscript (which it is not in the current version), starting the task with a low input weight makes sense.

We have modified the manuscript to explain these components of the model in the main text. In particular, we include sections explaining what “learning” means in the LDDM. First, we now explicitly describe and simulate a recurrent neural network from pixels, which provides a better account of the meaning of the input weights–taken literally, these could be synaptic weights from a population of neurons representing the visual input, to a population of neurons that integrate this signal, and gestures toward a mechanistic implementation in the brain. As the reviewer notes, though, these could in principle also be more abstract, and the model may describe functional connectivity implemented in more complex circuits.

“Within a trial, *N* dimensional inputs *s(t)*
∈RN arrive at discrete times *t* = 1*dt,* 2*dt,*∙∙∙ where *dt* is a small time step parameter. In our experimental task, *s(t)* might represent the activity of LGN neurons in response to a given visual stimulus. Because of eye motion and noise in the transduction from light intensity to visual activity, the response of individual neurons will only probabilistically relate to the correct answer at any given instant. In our simulations, we take *s(t)* to be the pixel values of the exact images presented to the animals, but transformed at each time point by small rotations (*±20deg*) and translations (*±25deg* of the image width and height), as depicted in Figure 3a. This input variability over time makes temporal integration valuable even in this visual classification task. To perform this integration, each input *s(t)* is filtered through perceptual weights *s(t)*
∈RN and added to a read-out node (decision variable) y^(t) along with i.i.d. η(t)∼N(0,co2dt). This integrator noise models internal neural noise. The evolution of the decision variable is given by the simple linear recurrence y^(t+dt)=y^(t)+ w(trial)⋅s(t)+ η(t) until the decision variable hits a threshold *±z(trial)* that is constant on each trial. Here the RNN already performs an integration through time (a choice motivated by prior experiments in rodents [37]), and improvements in performance come from adjusting the input-to-integrator weights *w(trial)* to better extract task relevant sensory information.”

In the second section, we describe the reduction of this RNN to the LDDM. The input weight in the reduction summarizes the functional effect of many individual weights in the RNN. In this sense, exactly as the reviewer says, the LDDM input weight is a more abstract quantity. However, it summarizes the behavior of the more mechanistically plausible weights of the RNN. We show that simulations of the reduction match those of a real RNN trained from pixels, verifying our methods.

“We start by noting that the input to the decision variable y^ at each time step is a weighted sum of many random variables, which by the law of large numbers will be approximately Gaussian. […] The dynamics of this “learning DDM” (LDDM) closely tracks simulated trajectories of the full network from pixels (Figure 3c-f blue trace, Figure 3—figure supplement 1; see Methods).”

Finally, a few choices made by the authors are critical for their findings, but are not sufficiently described in the main text. First, the observed reaction time (RT) is composed of a non-decision time (T0) and a decision time (DT). Experiments allow to measure RT, but the theory makes predictions about DT, which requires inferring T0. The magnitude of T0 impacts the results, but how it is inferred is currently buried in the Methods section. Second, the rodents sometimes choose to not immediately initiate a new trial (the "voluntary inter-trial interval"). The authors assume that rodents ignore this interval when maximizing reward rate, and find near-optimal reward rates under this assumption. Importantly, including this voluntary inter-trial interval makes reward rates drop significantly. However, these details are again buried in the Methods section, and are not mentioned nor discussed in the main text.

We added the following paragraph near the beginning of the Results section describing these parameters:

“Calculating mean normalized DT for comparison with the OPC requires knowing two quantities, DT and the average non-decision time per error trial *D_err_*. The average non-decision time *D_err_ = T_0_* +D¨err contains the motor and initial perceptual processing components of RT, denoted T0; and the post response timeout on error trials D¨err. Mean normalized DT is then the ratio DT/Derr. In order to determine each subject’s *DT*, we estimated T0 through a variety of methods, opting for a biological estimate (measured lickport latency response times and published visual processing latencies; Fig. 1–figure supplement 4). To ensure that our results did not depend on our choice of *T_0_*, we ran a sensitivity analysis on a wide range of possible values of *T_0_* (Fig. 1–figure supplement 4f). We then had to determine D¨err, which can contain mandatory and voluntary intertrial intervals. We found that the rats generally kept voluntary intertrial intervals to a minimum, and we interpreted longer intervals as effectively “exiting” the DDM framework (Fig. 1–figure supplement 5). As such, we defined D¨err to only contain mandatory intertrial intervals (see Methods, Fig. 1–figure supplement 6). To ensure that our results did not depend on either choice, we ran a sensitivity analysis on the combined effects of *T_0_* and a D¨err containing voluntary intertrial intervals on RR (Fig. 1–figure supplement 7). A full discussion of how these parameters were determined is included in the Methods.”

– The main conclusions hinge upon concurrent changes in both drift rate and decision boundary throughout learning. This change is assessed via fitting HDDM models. It is important that the authors fit and compare the following 3 DDM variants: (a) drift rate changes throughout learning but decision boundary remains fixed, (b) decision boundary changes throughout learning but drift rate remains fixed, (c) both drift rate and decision boundary change throughout learning.

We fit the 3 DDM variants suggested by the reviewers and found that models that allowed both drift and threshold to vary with learning provided the best fits. We discuss these model variants in the point below.

“Indeed, the best DDM model fits were those that allowed both threshold and drift rate to vary with learning, as was the case with the first stimuli the rats encountered, and in line with the LDDM model (Figure 4—figure supplement 1, Figure 4—figure supplement 2, Figure 4—figure supplement 3, Figure 4—figure supplement 4).”

– The HDDM fitting procedure is not fully described. In particular, it is not clear what parameters, asides the drift rate, the decision boundary and the non-decision time, varied. For instance, was the drift rate variability a free parameter? It is important to report more precisely the details of the model fits and, if simple variants of the DDM were used, to fit the data using more complex DDMs that include drift rate variability as a free parameter. Additionally, please report in Figure S5 all parameter estimates during learning and not just the drift rate and the decision boundary. Details around Equation (40) should also be in the main text. Furthermore, except for the internal noise, the setup looks very similar to that of Drugowitsch et al. (2019), and the relationship should be discussed somewhere in the main text.

In our original submission, we used the HDDM model to first fit a simple

DDM to 10 sessions of asymptotic behavior after learning stimulus pair 1 (n = 26) as a sanity check that a simple DDM was adequate (Figure 1—figure supplement 3). (We kept this figure unchanged).

To assess parameter changes during learning, in our original submission we fit separate, simple DDMs to each of the learning epochs of stimulus pair 1, and stimulus pair 2. As these were simple DDMs, drift rate variability was not a free parameter. For stimulus pair 1, the two learning epochs were the (1) first and (2) last 1000 trials for every subject. For stimulus pair 2, the three learning epochs were (1) 500 trials from a baseline session of stimulus pair 1 prior to introduction of stimulus pair 2, the (2) first 500 trials and the (3) last 500 trials of stimulus pair 2. Fewer trials were used because the stimulus pair 2 experiment had fewer sessions and trials overall. In order to succinctly address the modeling requests from the paper reviews, we modified the fitting procedure (described below) and replaced Figure S5 with new Figure 4—figure supplements 1-4, where we report the values of all parameters in every model fit.

The qualitative results were the same across all model fit variations: drift rate increased with learning and threshold decreased, even with the addition of drift rate variability. We report DICs (Figure 4—figure supplement 1) for each fit and arrive at two conclusions. First, models where both drift and threshold varied with learning fit the best, as opposed to models where drift and/or threshold were held constant. Second, more complex models involving drift rate variability did not provide a better fit over a simple DDM, and those that performed the best were those where drift and threshold were already allowed to vary with learning. This means that drift rate variability did not seem to provide a substantial benefit in explaining the data. Moreover, the direction in which drift rate variability changed with learning (increasing or decreasing with learning epoch) seemed to depend on which parameters also varied with learning (drift and/or threshold) leading to differing accounts. However, we cannot rule out that drift rate variability may indeed play a role in learning strategy. We leave the interpretability of such an account to our reviewers and our readers.

We have provided a more complete description of the revised HDDM fitting procedure in the Methods section:

“In order to verify that our behavioral data could be modeled as a drift-diffusion process, the data were fit with a hierarchical drift-diffusion model [121], permitting subsequent analysis (such as comparison to the optimal performance curve) based on the assumption of a drift-diffusion process. To verify that a DDM was appropriate for our data, we fit a simple DDM to 10 asymptotic sessions after learning stimulus pair 1 for *n* = 26 subjects (Fig. 1–figure supplement 3). In order to assess parameter changes across learning, we fit DDMs to the stimulus pair 1 experiment and the stimulus pair 2 experiment where the learning epochs were treated as conditions in each experiment. This allowed us to hold some parameters constant while conditioning others on learning. We fit both simple DDMs and DDMs with drift rate variability to the two experiments, allowing drift rate, threshold and drift rate variability to vary with learning epoch. In particular, we fit three broad types of models: (1) simple DDMs (Fig. 4–figure supplement 2), (2) DDMs + fixed drift rate variability (Fig. 4–figure supplement 3), and (3) DDMs + drift rate variability that varied freely with learning epoch (Fig. 4–figure supplement 4). For each of the types of models we held drift constant, threshold constant, or allowed both to vary with learning. The best fits, as determined by the deviance information criterion (DIC), came from models where we allowed both drift and threshold to vary with learning; the addition of drift rate variability did not appear to improve model fits (Fig. 4–figure supplement 1). For both learning experiments, drift rates increased and thresholds decreased by the end of learning, in agreement with previous findings [19, 53–57]. In addition, for the transparent stimuli experiment we fit a DDM model that allowed drift rate, threshold, drift rate variability and *T_0_* to vary with learning phase in order to observe the changes in drift rate and threshold 6–figure supplement 3.”

We included details around Equation 40 in the main text (Within-trial DriftDiffusion Dynamics subsection).

We now discuss Drugowitsch et al., 2019 at greater length in the discussion. In short, they treat learning in a Bayesian formulation, using primarily simulation-based methods, under the assumption that the diffusion boundaries (thresholds) are constant. By accounting for uncertainty in the weights, their Bayesian learning algorithm is more powerful than ours. The principal difference with our work, however, is the assumption of constant thresholds throughout learning. Our core interest is in how modification of response times (through changes in the decision threshold) impacts long term learning. Our simpler gradient descent learning model (still widely used throughout deep learning) permits analytical computation of average trajectories and updates that reveal the basic tradeoff between viewing time and learning speed in this setting. We believe future work could profitably combine Bayesian learning with threshold trajectories.

“Prior work in the DDM framework has investigated learning dynamics with a Bayesian update and constant thresholds across trials [35]. Our framework uses simpler error corrective learning rules, and focuses on how the decision threshold policy over many trials influences long-term learning dynamics and total reward. Future work could combine these approaches to understand how Bayesian updating on each trial would change long-term learning dynamics, and potentially, the optimality of different threshold strategies.”

– Currently the evolution of the mean RT during learning is examined. Plotting the change in the RT distribution (averaged/vincentised across participants) can be more informative about changes in the strategy being used than plotting the mean RT alone.

We have provided the change in correct RT distributions across learning in Figure 6—figure supplement 2. The change in RT distributions for stimulus pair 1 and 2 is qualitatively similar (slower RTs get faster, and faster RTs remain about the same) suggesting a similar mechanism (increase in drift accompanied by a decrease in threshold, which is indeed what we see in our DDM fits).

– The results of experiment 3 are interesting, but at the same time the behaviour in the transparent group requires more scrutiny. How do rats behave in this condition? It appears that random choice in combination with heuristic strategies (e.g. win-stay lose-switch) are viable possibilities. The argument that the SNR is intrinsically zero in this task. However, the signal in this experiment may contain non-zero, irrelevant information (such as choice history or feedback information). Plotting the RT distributions as a function of learning could provide insight in this regard, because boundary changes and SNR changes manifest differently in the shapes of RT distributions.

We have provided the change in vincentized correct RT distributions as a function of learning for the transparent stimuli experiment in Figure 6—figure supplement 2. The change in the RT distribution for transparent stimuli differs from that of stimulus pair 1 and 2 (slow and fast RTs all get faster), suggesting a different mechanism (decrease in threshold with a constant drift rate, as supported by our DDM fits in Figure 6—figure supplement 3). We added the following sentence to the text noting this: “Additionally, we considered the rats' entire *RT* distributions to investigate the effect of learnability beyond *RT* means. We found that while the *RT* distributions changed similarly from the beginning to end of learning for the learnable stimuli (stimulus pair 1 and 2), they differed for the unlearnable (transparent) stimuli, indicating an effect of learnability on the entire *RT* distributions (Figure 6—figure supplement 2). Hence rodents are capable of modulating their strategy depending on their learning prospects.”

The reviewers raise the point that the rats may be using non-stimulus information in order to sustain an SNR > 0. In fact, HDDM fits of this experiment (Figure 6—figure supplement 3) reveal that although drift rate does decrease with the transparent stimuli, it is still above 0. That being said, we observe a systematic decrease in threshold throughout learning with the transparent stimuli, in agreement with our model. This monotonic decrease in threshold is in contrast to the experiment with new visible stimuli (stimulus pair 2), where we also first observe a decrease in drift, but that decrease in drift is accompanied with an increase in threshold (Figure 4—figure supplements 2-4). After learning, threshold decays back to its baseline state, and drift also increases to match its baseline state (Figure 4—figure supplement 2-4).

To investigate whether the rats implemented stimulus-independent heuristic strategies in addition to random choice, we measured left/right bias and quantified the weights of bias, perseverance (choose the same port as previous trial) and win-stay/lose-switch (choose the port that was correct on the previous trial) in addition to the stimulus presented using a dynamic generalized linear model (PsyTrack: Roy et al., 2021; Figure 6—figure supplement 4). In general, bias seemed to increase with transparent stimuli in the direction that each individual was already biased during visible stimuli. The weights for perseverance and winstay/lose-switch also seemed to increase and fluctuate more during transparent stimuli, suggesting a greater reliance on these heuristics now that the stimulus was uninformative (stimulus weights dropped to 0, whereas they were strongly positive or negative during visible stimuli depending on each individual’s left/right stimulus mapping).

These results support the reviewers’ suggestion that the rats may indeed be relying more on stimulus-independent heuristic strategies. Although the rats may be sustaining a non-zero drift rate through stimulus-independent information, this might amount to a “monitoring” of the task in case any changes or reliably informative patterns do arise, rather than a strategy to improve perceptual learning as we argue is the case with the new visible stimuli.

We note that we still observe a monotonic decrease in threshold in our DDM fits, and a non-zero drift rate due to these heuristics does not contravene our conclusions.

Regarding heuristic strategies, we have added the following paragraph to the text:

“Although there is no informative signal in this task with transparent stimuli, the rats could still be using stimulus-independent signals, such as choice history or feedback, to drive heuristic strategies. Indeed, DDM fits indicated a non-zero drift rate even in the absence of informative stimuli (Figure 6—figure supplement 3 ). To investigate whether the rats implemented stimulus-independent heuristic strategies in addition to random choice, we measured left/right bias and quantified the weights of bias, perseverance (choose the same port as the previous trial) and win-stay/lose-switch (choose the port that was correct on the previous trial) [58]. In general, bias seemed to increase with transparent stimuli in the direction that each individual was already biased during visible stimuli. Perseverance and win-stay/lose-switch also seemed to increase and fluctuate more during transparent stimuli, suggesting a greater reliance on these heuristics now that the stimulus was uninformative (Figure 6—figure supplement 4). Engaging these heuristics may be a way that the rats expedited their choices in order to maximize *iRR* while still ``monitoring" the task for any potentially informative changes or patterns. Despite the fact that the animals' still engaged these non-optimal heuristics, the lack of learnability in the transparent stimuli still led to a change in strategy that was distinct from that with learnable stimuli.”

– The use of RNNs is not sufficiently supported beyond classical drift-diffusion modeling. The authors should either reframe the specific role of RNNs for supporting their key findings, or possibly remove them if they do not provide unique insights beyond classical drift-diffusion modeling.

Please see the response above to point #1.

– The authors should clarify early on in the manuscript what "learning" means in their model and theory, and why it makes sense to start with a low input weight (after describing the meaning of the input weight). The authors should also explain what T0 and D_RSI are, how they are determined (and the choices made to determine them), and how these parameters impact the results (also related to Figure S13). In particular the relationship between RT and DT is already required to understand Figure 1c, and many plots thereafter.

We added a better explanation of what “learning” means in the LDDM. We also explained T0 in more detail early in the Results section. We also simplified our timing terms to remove confusion around D_RSI.

Please see the full response to this comment above point #1.

– Box 1 provides some details of the model, but leaves out others – e.g., the different sources of noise in the model. From Box 1 alone, it is unclear how the asymptotic SNR or the iRR-sensitive threshold are computed. It is indeed nice that it is possible to derive Equation (3), but the equation itself is not particularly informative for the exposition of the main findings, and so it could be moved to the Methods section.

We replaced Box 1 with a more complete explanation of the model in the main text (Linear Drift-Diffusion Model (LDDM) section), including information on the asymptotic SNR and the threshold policies. We believe that including Equation 3 in the main text is important to demonstrate that the dynamics of learning in the RNN can be expressed as these specific dynamics for SNR in DDM terms, making it readily applicable to other settings. It is in our view a central result, not a method.